# SOUNDNESS AND COMPLETENESS: AN ALGORITHMIC PERSPECTIVE ON EVALUATION OF FEATURE ATTRIBUTION

## ABSTRACT

Feature attribution is a fundamental approach to explaining neural networks by quantifying the importance of input features for a model's prediction. Although a variety of feature attribution methods have been proposed, there is little consensus on the assessment of attribution methods. In this study, we empirically show the limitations of *order-based* and *model-retraining* metrics. To overcome the limitations and enable evaluation with higher granularity, we propose a novel method to evaluate the *completeness* and *soundness* of feature attribution methods. Our proposed evaluation metrics are mathematically grounded on algorithm theory and require no knowledge of "ground truth" informative features. We validate our proposed metrics by conducting experiments on synthetic and real-world datasets. Lastly, we use the proposed metrics to benchmark a wide range of feature attribution methods. Our evaluation results provide an innovative perspective on comparing feature attribution methods. Code is in the supplementary material.

## 1 INTRODUCTION

Explaining the prediction of machine learning (XML) models is an important component of trustworthy machine learning in various domains, such as medical diagnosis (Bernhardt et al., 2022; Khakzar et al., 2021b;a), drug discovery (Callaway, 2022; Jiménez-Luna et al., 2020), and autonomous driving (Kaya et al., 2022; Can et al., 2022). One fundamental approach to interpreting neural networks is feature attribution, which indicates how much each feature contributes to a model's prediction. However, different feature attribution methods can produce conflicting results for a given input (Krishna et al., 2022). In order to evaluate how well a feature attribution explains the prediction, different evaluation metrics have been proposed in the literature.

Despite valuable efforts and significant contributions to proposing a new evaluation strategy for feature attribution methods, several problems of concern remain. (1) Some evaluation strategies use duplicate or even conflicting definitions. For instance, (Ancona et al., 2018) define *sensitivity-n* as the equality between the sum of attribution and the output variation after removing the attributed features, while (Yeh et al., 2019) define the *sensitivity* as the impact of insignificant perturbations on the attribution result. (Lundstrom et al., 2022) provide another version of sensitivity, where the attribution of a feature should be zero if it does not contribute to the output. (2) The *retraining* on a modified dataset (Hooker et al., 2019; Zhou et al., 2022) is time-consuming. Furthermore, many retraining-based evaluations imply a strong assumption that only a part of the input is learned during retraining. For instance, ROAR (Hooker et al., 2019) assumes that a model only learns to use features that are not removed during perturbation, and (Zhou et al., 2022) assumes that a retrained model only learns watermarks added into the original images of a semi-natural dataset. Later in this work, we show that a model can pick up *any* features in the dataset during retraining. (3) Many evaluation metrics are only *order-sensitive*, meaning that they only evaluate whether one feature is more important than another while ignoring how differently the two features contribute to the output.

To overcome the discussed challenges, we propose to evaluate the alignment between attribution and informative features from an algorithmic perspective. The two proposed metrics work in conjunction with each other and reflect different characteristics of feature attribution methods. Our proposed metrics are both order-sensitive and value-sensitive, allowing for stricter differentiation between

two attribution methods. Therefore, the information provided by our metrics is more fine-grained compared to existing metrics. In addition, our proposed metrics can perform evaluation without knowing the "ground truth" informative features for model inference, as we utilize the model's performance as an approximate metric to compare different feature attribution methods.

To summarize our contributions:

- We empirically reveal the limitations of existing evaluation strategies for feature attribution. We show that order-based evaluations can be underperforming. We further demonstrate that evaluations with retraining are not guaranteed to be correct.

- We draw inspiration from algorithm theory and propose two novel metrics for faithfully evaluating feature attribution methods. Our approach requires no prior knowledge about "ground truth" informative features. We conduct extensive experiments to validate the correctness of our metrics and empirically show that our approach overcomes the problems associated with existing metrics.

- We comprehensively benchmark feature attribution methods with our proposed metrics. Our benchmark reveals some undiscovered properties of existing feature attribution methods. We also examine the effectiveness of ensemble methods to showcase that our metrics can help to create better feature attribution methods.

## 2 RELATED WORK

**Feature Attribution**  Attribution methods explain a model by identifying informative input features given an associated output. Gradient-based methods (Simonyan et al., 2014; Baehrens et al., 2010; Springenberg et al., 2015; Zhang et al., 2018; Shrikumar et al., 2017) produce attribution based on variants of back-propagation rules. Another family of methods attempts to approximate Shapley values (Shapley, 1997), where features are considered as cooperative players with different contributions. Examples of these methods include DeepSHAP (Lundberg & Lee, 2017) and Integrated Gradients (IG) (Sundararajan et al., 2017; Sundararajan & Najmi, 2020; Lundstrom et al., 2022). Additionally, perturbation-based methods (Fong & Vedaldi, 2017; Fong et al., 2019; Ribeiro et al., 2016) such as Extremal Perturbations (ExPeturb) (Fong et al., 2019) rely on perturbing input features and measuring the impact on output. Moreover, IBA (Schulz et al., 2020) and InputIBA (Zhang et al., 2021) are derived from information bottlenecks. InputIBA finds a better prior for the information bottleneck at the input, thus producing more fine-grained attribution maps than IBA. Lastly, attribution methods based on activation maps like CAM (Zhou et al., 2016) and GradCAM (Selvaraju et al., 2017) use activations or gradients of hidden layers.

In this work, we analyze and study some representatives from each of these categories. Specifically, we assess GradCAM (Selvaraju et al., 2017), DeepSHAP (Lundberg & Lee, 2017), IG (Sundararajan et al., 2017), ExPerturb (Fong et al., 2019), IBA (Schulz et al., 2020) and InputIBA (Zhang et al., 2021).

**Evaluation metrics for feature attribution**  Earlier efforts have attempted to evaluate various feature attribution methods. One category of these methods is *expert-grounded metrics* that rely on visual inspection (Yang et al., 2022), pointing game (Zhang et al., 2018), or human-AI collaborative tasks (Nguyen et al., 2021). However, the evaluation outcome is subjective and does not guarantee consistency. Another category is *functional-grounded metrics* (Petsiuk et al., 2018; Samek et al., 2016; Ancona et al., 2018) that perturb input according to *attribution order* and measure the change of model's output. Prior works usually consider two removal orders: MoRF (Most Relevant First) and LeRF (Least Relevant First). (Hooker et al., 2019) argued that removing features can lead to adversarial effects. They proposed a method ROAR to retrain the model on the perturbed dataset to mitigate the adversarial effect. Later, ROAD (Rong et al., 2022) showed that the perturbation masks can leak class information, leading to an overestimated evaluation outcome. Recently, (Zhou et al., 2022) proposed to inject "ground truth" features into the training dataset and forced the model to learn these features solely. They then tested if an attribution method identified these features. Furthermore, (Khakzar et al., 2022) propose to generate datasets with *null features* and then test the axioms for feature attribution. Another sub-type in functional-grounded metrics focuses on sanity checks for attribution methods (Adebayo et al., 2018). The key differences between our approach

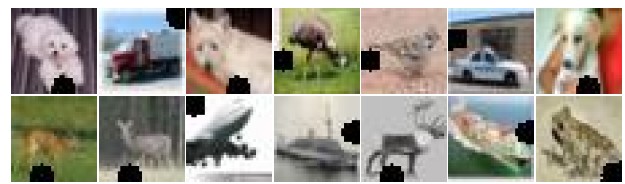 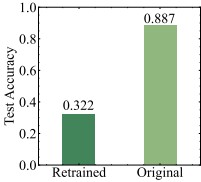

(a) Example images in the modified dataset for retraining (b) Accuracy

Figure 1: Retraining on the modified dataset. Our malicious modification scheme perturbs a small fraction of background pixels but introduces a high correlation with classes. The test accuracy on the unperturbed test set shows that the retrained model ignores the remaining features but learns to mostly rely on introduced artifacts for classification. Hence, the low test accuracy here is not due to information removal, but due to not learning from the true class-related information.

and the aforementioned approaches are: (1) our approach does not entail retraining on modified or semi-natural datasets, and (2) our approach can distinguish between attribution methods that arrange features in the same order but assign features with different attribution values.

## 3 PROBLEM FORMULATION AND METHOD

### 3.1 ISSUES WITH FEATURE IMPORTANCE EVALUATIONS

In this section, we illustrate two complications with feature importance evaluations. One issue is related to order-based evaluation. Another issue is related to evaluations that rely on retrained models. We discuss these issues in detail in the following text.

So far, many feature importance evaluations operate by removing features by attribution order, then observing the variation in the classifier's output. In this work, we argue that order-based evaluations are insufficient. Since these evaluations perturb features based on their relative order of attribution values, the magnitude of attribution values is less considered during evaluation. Subsequently, different attribution methods can have similar evaluation outcomes as long as the relative attribution order is preserved regardless of the actual attribution value. We empirically demonstrate this conflict with one falsification experiment shown in Fig. 3c left.

Another issue with regard to the evaluation of feature attribution methods is retraining. Some evaluation approaches use retraining on a purposefully modified dataset, then report the performance of the retrained model. Retraining-based evaluations assume either the retrained model learns solely from remaining features after perturbation, or the retrained model exclusively relies on introduced features during perturbation. However, neither of these assumptions necessarily holds, as there is no hard constraint in objective functions or learning procedures to restrain a model from learning certain features. If the retraining is not ensured to follow these assumptions, the model's performance is insufficiently trustworthy for evaluating feature attribution methods.

We again demonstrate the issue related to retraining using one counterexample. Since retraining is usually conducted on a modified dataset, we maliciously design a modification scheme to show that the model cannot learn the remaining features in this case. Our malicious strategy is to perturb a small number of pixels from the training data, such that the foreground object is intact in the image, but the perturbed pixels are now highly correlated to the class. Specifically, we choose CIFAR-10 (Krizhevsky et al., 2009a) as the original dataset and perturb $5\%$ of the pixels for each image in the training set. As objects are usually centered in CIFAR-10 images, we only perturb pixels close to the edges. However, the perturbation is class-related. If an image belongs to a certain class, we always remove pixels at a fixed position for that class. Fig. 1a shows examples of perturbed images. We use the modified dataset for retraining and report the accuracy on the unperturbed test set to evaluate how many remaining features are learned during retraining. More details of this experiment can be found in Appendix E. Fig. 1b shows test accuracy of the original and retrained model. We observe that the retrained model has not learned to classify images using the remaining features. We hypothesize that modifying a dataset changes the loss landscape, and the model would converge to another optimum that has steeper gradients in the vicinity. Nevertheless, this example

shows that it is difficult to constrain the learning process. Subsequently, a retrained model can learn to rely on any features in the retraining dataset to make predictions, regardless of whether the feature is introduced by modification or from the original dataset. Many retraining-based evaluations assume that a model learns a specific part of the input and define evaluation metrics based on this assumption. Since we showed this assumption can be violated easily, these evaluations based on retraining have no guarantee of correctness.

(Zhou et al., 2022) recently proposed an evaluation method that modifies a dataset by adding artificial ground truth, then evaluates feature attribution methods on the model trained on the modified dataset. Nevertheless, training on a modified semi-natural dataset also comes with a problem. Specifically, the learning task deviates from the original learning task when training on a semi-natural dataset. As a result, explanation methods can behave very differently on the semi-natural dataset, and we cannot obtain the actual performance of explanation methods. We demonstrate the issue by evaluating the performance of models as well as explanation methods on three different datasets. Here, we use CIFAR-100 (Krizhevsky et al., 2009b) as our learning task, then we add watermarks to inject artificial ground truth as (Zhou et al., 2022) suggested. Moreover, we generate a pure synthetic dataset having only watermarks in images. After training models on three datasets, we obtain test accuracy of $70.4\%$ on CIFAR-100, $99.0\%$ on the semi-natural dataset, and $99.6\%$ on the pure synthetic dataset. This shows that the difficulty of the original task is significantly greater than the task associated with the semi-natural dataset. However, *we want to evaluate the performance of explanation methods on complex models trained on real-world datasets*. An over-simplified dataset deviating from real datasets only creates a primitive evaluation setup. More importantly, as shown in Appendix B, GradCAM outperforms IG and DeepSHAP on CIFAR-100, while IG and DeepSHAP are much better than GradCAM on the semi-natural and pure synthetic datasets. This suggests that evaluating attribution methods across different datasets are inconsistent and may lead to unfaithful conclusions.

## 3.2 An algorithmic perspective evaluation

In Section 3.1, we discussed the issue with order-based evaluations. To overcome the limitations mentioned above and provide evaluation results with higher granularity, we develop an evaluation based on algorithm theory. Specifically, since feature attribution methods are algorithms, we can apply metrics in algorithm theory to assess their correctness. The correctness of an algorithm is typically related to two concepts: *soundness* and *completeness* (Smith, 2010). While prior works about attribution method evaluations mainly focus on measuring the relative correctness of a method, they ignore the fact that violation of either soundness or completeness can result in a partially correct algorithm. As a first step, we adapt the definitions from algorithm theory to formally define complete and sound feature attribution methods, respectively. Given a set $\mathcal{F}$ that contains all features in the dataset, let $\mathcal{A} \subseteq \mathcal{F}$ be a set of salient features identified by an attribution method, and given a set of informative features $\mathcal{I} \subseteq \mathcal{F}$ for the model, we have:

**Definition 3.1** (Complete Feature Attribution). *An attribution method is complete iff all informative features for the model are identified by the attribution method. $\forall F \in \mathcal{I}, F \in \mathcal{A}$.*

**Definition 3.2** (Sound Feature Attribution). *An attribution method is sound iff all salient features by the attribution method are informative for the model. $\forall F \in \mathcal{A}, F \in \mathcal{I}$.*

However, almost all existing attribution methods are neither sound nor complete. In this case, evaluating whether a method is sound or complete provides very limited information and cannot be used as a comparative metric. To measure the relative deviation from being perfectly sound and complete, we define soundness and completeness as more relaxed metrics. Completeness assesses the performance of an attribution method by checking how well the resulting attribution map covers the informative features. Complementary to completeness, soundness evaluates how well salient features contain true informative features. Formally, we define completeness as $|\mathcal{A} \cap \mathcal{I}|/|\mathcal{I}|$, and soundness as $|\mathcal{A} \cap \mathcal{I}|/|\mathcal{A}|$. The operator $|\cdot|$ calculates the set volume. Unlike conventional cardinality, which counts the number of features, we define the volume as the total attribution or information in a set. By using $|\cdot|$, we can obtain the overall effect of features with different amounts of information or with different attribution values.

There are three major challenges in evaluating soundness and completeness. One challenge is that feature importance and feature attribution are not discrete, so we must evaluate them at different

value levels. Another harder challenge is that we have no information about informative features $\mathcal{I}$. Therefore, it is intractable to directly calculate $|\mathcal{A} \cap \mathcal{I}|$. Lastly, we cannot retrain a model or modify the dataset during evaluation due to the above-illustrated issue with retraining. We show in subsequent sections how we address these challenges and evaluate completeness and soundness.

### 3.2.1 COMPLETENESS EVALUATION

As discussed, completeness evaluates if all informative features are included in attribution maps. Although we have no knowledge about informative features for the model, we can leverage the model to provide indirect indications. We make the following assumption throughout the work:

**Assumption 1.** *Given a dataset $\mathbb{D}$, $\mathcal{F}$ be the set of all input features in $\mathbb{D}$, $f$ be a model, and $\rho$ be a performance metric to assess the performance of model $f$. $\forall \mathcal{F}_1, \mathcal{F}_2 \subseteq \mathcal{F}$, if $\rho(f(\mathcal{F}_1)) < \rho(f(\mathcal{F}_2))$, then $|\mathcal{I} \cap \mathcal{F}_1| < |\mathcal{I} \cap \mathcal{F}_2|$, where $\mathcal{I} \subseteq \mathcal{F}$ is the set of informative features for model $f$.*

According to Definition 3.1, removing salient features generated from a complete method would also cause the removal of informative features. We can leverage this statement to compare the completeness across different attribution methods.

**Theorem 3.1.** *Assume two attribution methods with returned salient features $\mathcal{A}_1$ and $\mathcal{A}_2$, respectively. If $\rho(f(\mathcal{F} \setminus \mathcal{A}_1)) < \rho(f(\mathcal{F} \setminus \mathcal{A}_2))$, then the attribution method associated with $\mathcal{A}_1$ is more complete than the one associated with $\mathcal{A}_2$.*

**Proof**: Assume salient features to be $\mathcal{A}$. After removing these features from the input, the set of remaining informative features is $\mathcal{I} \setminus \mathcal{A}$. Then, we have:

$$\frac{|\mathcal{I} \setminus \mathcal{A}|}{|\mathcal{I}|} = \frac{|\mathcal{I} \setminus (\mathcal{A} \cap \mathcal{I})|}{|\mathcal{I}|} = 1 - \frac{|\mathcal{A} \cap \mathcal{I}|}{|\mathcal{I}|} := 1 - completeness(\mathcal{A}). \tag{1}$$

Because our defined $|\cdot|$ satisfies associativity, we have $|\mathcal{I} \setminus (\mathcal{A} \cap \mathcal{I})| = |\mathcal{I}| - |\mathcal{A} \cap \mathcal{I}|$ in the above equation. In addition, since the set of informative features $\mathcal{I}$ is constant given a fixed model, we can ignore $|\mathcal{I}|$ and solely use $|\mathcal{I} \setminus \mathcal{A}|$ to evaluate relative completeness. However, $|\mathcal{I} \setminus \mathcal{A}|$ is not accessible. To approximate $|\mathcal{I} \setminus \mathcal{A}|$, we leverage Assumption 1 and pass $\mathcal{F} \setminus \mathcal{A}$ through the target model $f$ to report the performance measure $\rho(f(\mathcal{F} \setminus \mathcal{A}))$. If $\rho(f(\mathcal{F} \setminus \mathcal{A}_1)) < \rho(f(\mathcal{F} \setminus \mathcal{A}_2))$, then $|\mathcal{I} \setminus \mathcal{A}_1| < |\mathcal{I} \setminus \mathcal{A}_2|$. Hence, the completeness of $\mathcal{A}_1$ is larger than the completeness of $\mathcal{A}_2$. We have proved Theorem 3.1.

Based on the above analysis, we briefly describe our procedure for evaluating completeness at different attribution thresholds. We start with the unperturbed input and gradually remove input features above the designated attribution values. Then we pass the remaining features to the model and report the model performance difference between the origin and remaining features. A higher score difference means higher completeness. As noted in ROAD, imputation can mitigate the class information leakage and adversarial effect. Therefore, we impute perturbed images to diminish the undesired side effects caused by perturbation. The detailed procedure is shown in Appendix F.

### 3.2.2 SOUNDNESS EVALUATION

After we defined a metric to evaluate the completeness, we propose a metric for soundness evaluation. We adopt the same notation from Section 3.2.1 and use $\mathcal{A}$ for a set of salient features. We further define $\mathcal{I}_v \subseteq \mathcal{I}$ such that $|\mathcal{I}_v| = v$. The value $v$ is an abstract notion denoting the amount of useful information for model prediction. We refer to this value $v$ as the *information level*. Nevertheless, $v$ is intractable in our setting. Based on Assumption 1, if two sets of features can yield the same model performance, then the information level of these two sets of features is identical. Subsequently, *we use model performance as an indirect indicator for information level and measure the soundness at different model performance levels*. To have a unique $\mathcal{I}_v$, we further define the features belonging to $\mathcal{I}_v$ as *the most salient informative features*, i.e., every feature in $\mathcal{I}_v$ is informative for decision making and has higher attribution value compared with other informative features. To tackle the continuity of attribution values, we define soundness at information level $v$ as Soundness $= |\mathcal{A} \cap \mathcal{I}_v| / |\mathcal{A}|$. Similarly to completeness evaluation, $|\mathcal{A} \cap \mathcal{I}_v|$ is infeasible to calculate, since we have no knowledge of informative features. Nevertheless, we can leverage the model to conduct soundness evaluation without calculating $|\mathcal{A} \cap \mathcal{I}_v|$. We show this in the following analysis:

**Lemma 3.2.** *Let $\mathcal{A}_i, \mathcal{A}_j$ be two sets of salient features returned from different attribution methods, and $\mathcal{I}_v$ be informative features at information level $v$, if $\rho(f(\mathcal{A}_i)) = \rho(f(\mathcal{A}_j)) = \rho(f(\mathcal{I}_v))$, then $|\mathcal{A}_i \cap \mathcal{I}| = |\mathcal{A}_j \cap \mathcal{I}| = |\mathcal{I}_v|$.*

The proof of Lemma 3.2 is shown in Appendix C. With the help of Lemma 3.2, we can find $\mathcal{A}$ that satisfies $|\mathcal{A} \cap \mathcal{I}| = |\mathcal{I}_v|$. However, we still lack the necessary information to obtain $\mathcal{I}_v$. One issue is that the resulting $\mathcal{A}$ is an overestimate of $\mathcal{I}_v$.

**Lemma 3.3.** *Given a set of features $\mathcal{A}$ and a set of informative features $\mathcal{I}_v$, if $\rho(f(\mathcal{A})) = \rho(f(\mathcal{I}_v))$, then we have $\mathcal{I}_v \subseteq \mathcal{A}$.*

The proof of Lemma 3.3 is shown in Appendix D. The proof of Lemma 3.3 implies that if we can exclude non-informative features in $\mathcal{A}$, we would obtain $|\mathcal{I}_v|$ without overestimation. Thus, we can obtain the soundness at information level $v$ without explicitly calculating $|\mathcal{A} \cap \mathcal{I}_v|$:

**Theorem 3.4.** *If $\exists \mathcal{A}^* \subseteq \mathcal{A}$, such that $\rho(f(\mathcal{I}_v)) = \rho(f(\mathcal{A}^*))$, and $\mathcal{A}^* \cap (\mathcal{F} \setminus \mathcal{I}) = \emptyset$, then the soundness at information level $v$ is $\frac{|\mathcal{A}^*|}{|\mathcal{A}|}$.*

**Proof**: For $\mathcal{A}^* \subseteq \mathcal{A}$ that satisfies $\rho(f(\mathcal{I}_v)) = \rho(f(\mathcal{A}^*))$, we have $|\mathcal{A}^* \cap I| = |\mathcal{I}_v|$. Furthermore, if $\mathcal{A}^* \cap (\mathcal{F} \setminus \mathcal{I}) = \emptyset$, then $\mathcal{A}^* \subseteq \mathcal{I}$. Since $|\mathcal{A}^* \cap \mathcal{I}| = |\mathcal{I}_v|$ and $\mathcal{A}^* \subseteq \mathcal{I}$, we have $|\mathcal{A}^*| = |\mathcal{I}_v|$.

Theorem 3.4 suggests a way to evaluate soundness. Instead of calculating $\mathcal{A} \cap \mathcal{I}$ for any possible choice of $\mathcal{A}$, we can gradually increase $\mathcal{A}$ by adding a small fraction of salient features $\Delta \mathcal{A}$ into $\mathcal{A}$. If adding $\Delta \mathcal{A}$ has no positive effect on model performance, i.e., $\rho(f(\mathcal{A} + \Delta \mathcal{A})) \leq \rho(f(\mathcal{A}))$, we would skip these features and continue adding other salient features. Once we include enough salient features and find a $\mathcal{A}^*$ that satisfies $\mathcal{A}^* \cap \mathcal{I} = \mathcal{I}_v$, we calculate $\frac{|\mathcal{A}_*|}{|\mathcal{A}|}$ as the soundness at information level $v$.

In practice, we first gradually include the most salient features into the input. Then, we examine and exclude non-informative features based on the change in model performance. At each step, we compute the *attribution ratio*, i.e., the ratio of informative attribution divided by the total attribution. We then plot the curve of attribution ratio vs. model's performance. For a fixed performance score (equivalent to information level), a higher attribution ratio indicates a better attribution method. We also apply image imputation here as in Section 3.2.1. The details are shown in Appendix G.

# 4 EXPERIMENTS

Unless specified otherwise in the remaining text, we use an ImageNet (Deng et al., 2009) pre-trained VGG16 (Simonyan & Zisserman, 2015) as the classifier and perform feature attribution on the ImageNet validation set. We follow the same linear imputation strategy described in ROAD (Rong et al., 2022) to mitigate the class information leakage or adversarial effect. More details on experiment configurations can be found in Appendix H.

## 4.1 VALIDATION TESTS ON COMPLETENESS AND SOUNDNESS EVALUATIONS

In addition to our analysis in previous sections, the incentive of this section is to empirically validate the correctness of our proposed soundness and completeness metrics. Prior works on feature attribution evaluation focus more on the definition and heuristic of evaluation strategies but ignore testing on real and adversarial cases. Subsequently, many proposed heuristics were later discovered to have flaws. Therefore, validation is essential for evaluation metrics.

Our validation experiments have two complementary settings. We first use a synthetic dataset and design a model to validate our evaluation metrics. Because the model is fully transparent, we can simply obtain the ground truth features for the model. Hence, we can thoroughly examine whether our proposed evaluations have the desired behavior in different scenarios. Then, we conduct our experiment on a real, complex dataset to show the behavior of our metrics in a more realistic setting.

### 4.1.1 SYNTHETIC DATASET

In this experiment, our goal is to design a synthetic dataset and a simple linear model for feature attribution. With a synthetic dataset and a primitive model, we can perform feature attribution to

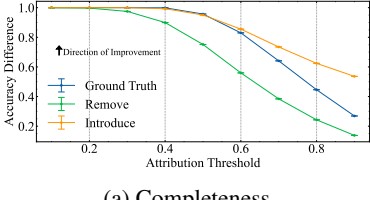

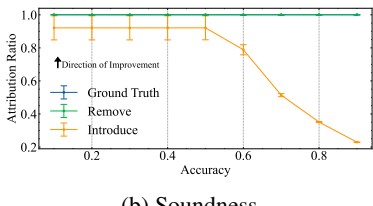

(a) Completeness
(b) Soundness

Figure 2: Completeness and soundness evaluations on the synthetic dataset. Attribution threshold $t$ indicates that pixels with attribution greater than $t$ are masked. The curve and error bar, respectively, show the mean and variance of 1000 trials of random attribution removal and introduction. The curves of Remove and Ground Truth are overlapped in (b) as they both saturate at 1. Both soundness and completeness evaluations can faithfully reflect the changes we made in attribution maps.

obtain "ground truth" attribution maps. Given ground truth attribution maps that are already sound and complete, we can apply modifications to them and anticipate the changes in completeness or soundness. Specifically, increasing attribution values of non-informative features in ground truth attribution maps would hurt the soundness of the resulting maps but improve completeness. In contrast, removing attribution from ground truth attribution maps would reduce completeness but not soundness. Hence, we can use our proposed soundness and completeness metrics to evaluate the modified attribution maps and check if the evaluation outcome can reflect the changes in soundness and completeness after modification.

The synthetic dataset is a two-class dataset. Data points are sampled from a 200-dimensional Gaussian distribution $\mathcal{N}(\mathbf{0}, \mathbf{I})$. If all features from a data point sum up to be greater than zero, we assign a positive label to the data point, and otherwise, we assign a negative label. The model is designed to fully replicate the data generation process and can be formulated as $y = \sigma(\sum_i x_i)$, where $x_i$ is the $i$-th feature and $\sigma(\cdot)$ is a step function that rises at $x = 0$. As the model is a transparent linear model, we can obtain the sound and complete "ground truth" attribution maps. More details on the attribution process are described in Appendix H.3. We randomly add and remove attribution from "ground truth" feature maps 1000 times each, producing 2000 different modifications, and then compare the soundness and completeness between modified and original attribution maps using our metrics.

Based on the statistical results shown in Fig. 2, we observe that the completeness of *Remove* is consistently higher than the completeness of the ground truth attribution, while the completeness of *Introduce* is always lower than the completeness of the ground truth attribution. In soundness evaluation, we observe that these three methods' ranking is reversed. Note that the soundness of the ground truth attribution is the optimum state of 1, and hence, Remove can also only reach this state. In conclusion, the behavior of soundness and completeness evaluations are as expected, thus validating that our proposal is correct in this case.

### 4.1.2 IMAGENET

Unlike the experiment in Section 4.1.1 (for which sound and complete attribution maps can be easily obtained), the ground truth attribution maps for ImageNet are inaccessible. Therefore, we cannot guarantee that the introduced attribution is exclusively added to non-informative features. In other words, the introduced attribution does not necessarily reduce soundness. Due to this natural constraint, we herein only compare the completeness between modified and original attribution maps. Here, we modify the GradCAM attribution maps in this experiment. We expect modified attribution maps with removal operation to be less complete than GradCAM, since the removal operation only removes attribution. On the other hand, adding attribution on GradCAM attribution maps would result in more complete attribution maps than GradCAM attribution maps.

To cover various modification cases, we implement three pairs of different schemes to modify the attribution maps obtained on ImageNet. When removing attribution, *Constant* and *Random* lower attribution by a constant and random positive value, respectively; *Partial* sets a part of the pixels' attribution to 0. When introducing additional attribution, *Constant* and *Random* raise attribution by a constant and random positive value, respectively; *Partial* sets the attribution of some less important pixels to a high value. Some examples are visualized in Fig. 3. The details of attribution modifica-

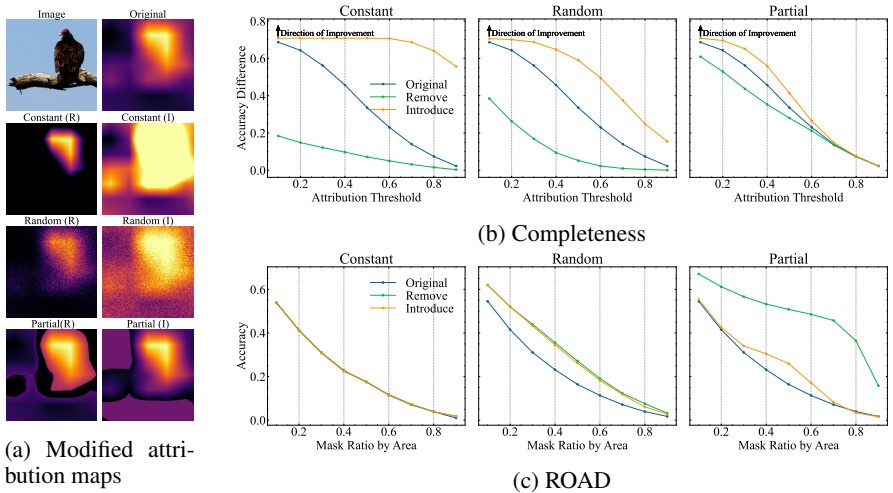

(a) Modified attribution maps

(b) Completeness

(c) ROAD

Figure 3: Comparison between modified and original (GradCAM) attribution maps in terms of completeness. In (a), modifications *Remove* and *Introduce* are denoted by (R) and (I), respectively. From (b), we can see that completeness evaluation is able to correctly capture the changes caused by removing and introducing attribution. However, ROAD (c) cannot distinguish between these changes in *Constant* and *Random* modifications. Further examples are shown in Appendix H.3.

tion can be found in Appendix H.3. The results in Fig. 3 are consistent across different modification schemes and conform to our expectations. In summary, the experiment results validate the correctness of our proposed metrics. Furthermore, as depicted in Fig. 3, ROAD fails to distinguish between the modified attribution maps in *Constant* and *Random* modification schemes. This shows that our completeness metric can effectively overcome the limitation of order-based evaluation metrics.

## 4.2 BENCHMARKING ATTRIBUTION METHODS

The ultimate purpose of developing metrics is to use them to compare attribution methods. Therefore, we benchmark some representative attribution methods here and hope the empirical findings and implications can inspire future research in feature attribution. In Fig. 4, we first show a sample to provide readers with an intuition about the soundness and completeness of these attribution methods. The benchmark results are presented in Fig. 5a and Fig. 5b. Several observations can be made as follows:

**Complete methods are not necessarily sound (and vice versa).** For instance, ExPerturb appears far more complete than other competitors, owing to the fact that the perturbed area is large enough to cover foreground objects. Nevertheless, ExPerturb is unsound, since it assigns high attribution to background pixels in the perturbed area. Visual results in Fig. 4 support our argument. In addition, two equally complete methods can exhibit different levels of soundness (and vice versa), as exemplified in the comparison of IG vs. DeepSHAP shown in Fig. 5a and Fig. 5b.

**Layer attribution methods outperform axiomatic ones.** *Layer attribution methods* like GradCAM and IBA perform attribution on a hidden layer and then interpolate the attribution map to the input size. InputIBA also belongs to this category, as it leverages the distribution on a hidden layer. In contrast, axiomatic methods such as DeepSHAP and IG perform attribution directly on input pixels. These methods usually produce very sparse attribution maps, which explains why they are less complete than layer attribution methods. The interpolation operation also enhances the completeness of layer attribution methods. One might expect layer attribution's proclivity for assigning attribution to the background in interpolated attribution maps. However, layer attribution methods appear even sounder than axiomatic ones in our metric: We first emphasize that IG (or DeepSHAP) also assigns small but non-zero attribution values to background pixels. The attribution of background will be determined as false attribution. Furthermore, IG's total amount of attribution is substantially smaller than layer attribution methods. Therefore, the false attribution in IG leads to a more considerable drop in the attribution ratio – hence, the soundness. The observation also implies that layer

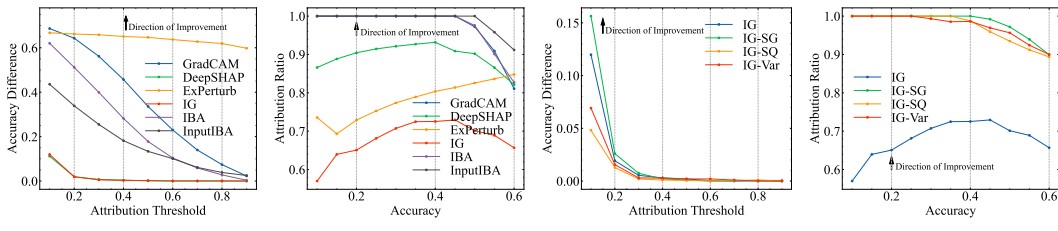

Figure 4: Sample attribution maps. We observe clear inconsistency between attribution methods. The image is randomly chosen, and more examples are provided in Appendix H.2.

(a) Completeness of various attribution methods

(b) Soundness of various attribution methods

(c) Completeness of IG ensembles

(d) Soundness of IG ensembles

Figure 5: The completeness and soundness benchmarks. (a)-(b) Comparison between various attribution methods. A method that shows high completeness does not necessarily imply that it also yields high soundness. Layer attribution methods such as IBA, GradCAM are not only more complete but also sounder than DeepSHAP and IG. (c)-(d) Comparison between IG ensembles. Although ensemble methods improve soundness, not all of them improve completeness.

attribution methods exhibit more distinct differentiation between foreground and background. This property may benefit downstream tasks, such as weakly-supervised object localization (Zhou et al., 2016; Selvaraju et al., 2017).

### 4.3 EVALUATING ENSEMBLE METHODS

Several ensemble methods (Smilkov et al., 2017; Hooker et al., 2019; Adebayo et al., 2018) have been recently proposed to enhance the performance of IG and DeepSHAP. In this experiment, we evaluate ensemble methods in order to investigate whether applying ensemble tricks is beneficial. By doing so, we demonstrate that our evaluation metrics can be used as a guide for improving feature attribution methods. We consider three ensemble methods of IG: SmoothGrad (IG-SG) (Smilkov et al., 2017), SmoothGrad[2] (IG-SQ) (Hooker et al., 2019), and VarGrad (IG-Var) (Adebayo et al., 2018). The benchmark results are presented in Fig. 5c and Fig. 5d.

Surprisingly, only one method IG-SG improves completeness. The completeness results are consistent with the visual results shown in prior work (we also present visual results in Appendix H.4 for further examination). Compared to IG, IG-SG assigns attribution to a larger set of pixels. Therefore, IG-SQ and IG-Var have less coverage of informative features. In addition, all three ensemble methods improve the soundness of IG to be on par with GradCAM and IBA. As noted in (Smilkov et al., 2017), the gradients fluctuate dramatically in neighboring samples. Hence, aggregating the attribution of neighboring samples can reduce false attribution and enhance soundness.

## 5 CONCLUSION

In this work, we first revealed the issues with order-based and model-retraining approaches. We then introduced a methodology to evaluate the soundness and completeness of feature attribution methods based on algorithm theory. Compared to existing metrics, our metrics have higher differentiation granularity. We circumvented the need to know "ground truth" informative features by utilizing the model's performance as an approximate and comparative assessment. Validation experiments provided convincing evidence for the effectiveness of our metrics. We also benchmarked various attribution methods and demonstrated that our metrics contribute towards a better understanding of feature attribution and can benefit future research on explainability.

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

## 6 APPENDIX

## A NOTATIONS

Table 1 lists the notations used in this paper.

Table 1: Table of notations.

| | |
|---|---|
| $\mathcal{F}$ | a set that contains all features in the dataset |
| $\mathcal{A}$ | a set of attributed features |
| $\mathcal{I}$ | a set of informative features for the model |
| $\lvert \cdot \rvert$ | the operator to calculate the set volume |
| $F$ | a single feature in $\mathcal{F}$ |
| $v$ | a scalar |
| $f$ | a model to be explained |
| $\rho$ | a performance metric to assess the performance of model $f$ |
| $\mathbb{D}$ | a (labeled) dataset |

## B EVALUATING ATTRIBUTION METHODS ON RETRAINED MODELS

In Section 3.1, we argue that the attribution methods can behave much differently when explaining the model retrained on a semi-natural dataset. As a result, it is not faithful to use the evaluation result on the semi-natural dataset as an assessment for the feature attribution methods. In this section, we demonstrate this issue with an experiment.

We set up the experiment as follows: we first re-assign the labels for CIFAR-100 (Krizhevsky et al., 2009b) images and add watermarks as suggested in (Zhou et al., 2022), obtaining a semi-natural dataset. Next, we inject watermarks to a blank canvas, obtaining a pure-synthetic dataset. After that, we train a VGG-16 on the CIFAR-100, semi-natural, and pure synthetic dataset, respectively. After obtaining the classification models, we apply GradCAM, IG, and DeepSHAP to them to get the attribution maps. In the end, we benchmark the three attribution methods on each dataset using ROAD (Rong et al., 2022).

The models achieve $70.4\%$ on CIFAR-100, $99.0\%$ on the semi-natural dataset, and $99.6\%$ on the pure synthetic dataset, respectively. The difference of accuracy shows that the learning tasks are differently complex across three datasets. Futhermore, as depicted in Fig 6, GradCAM outperforms IG and DeepSHAP on CIFAR-100, while IG and DeepSHAP are much better than GradCAM on the semi-natural and pure synthetic datasets. *The evaluation results on the semi-natural dataset cannot correctly reflect attribution methods' performance on the real-world dataset.*

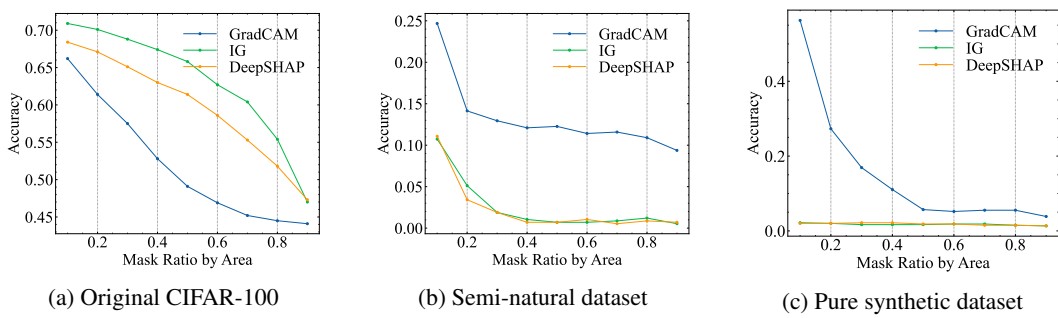

(a) Original CIFAR-100      (b) Semi-natural dataset      (c) Pure synthetic dataset

Figure 6: ROAD evaluation on different datasets. The performance of attribution methods can be much different between the real-world dataset and semi-natural (or pure synthetic) dataset.

## C PROOF OF LEMMA 3.2

Here, we show the Lemma 3.2 again to avoid frequent turning pages for readers.

**Lemma.** $\forall \mathcal{A}_i, \mathcal{A}_j$ *as salient features returned from different attribution methods, and* $\forall \mathcal{I}_v$ *informative features at information level* $v$*, if* $\rho(f(\mathcal{A}_i)) = \rho(f(\mathcal{A}_j)) = \rho(f(\mathcal{I}_v))$*, then* $\mathcal{A}_i \cap \mathcal{I} = \mathcal{A}_j \cap \mathcal{I} = \mathcal{I}_v$*.*

**Proof:** we use result from Assumption 1. Since $\rho(f(\mathcal{A}_i)) = \rho(f(\mathcal{A}_j)) = \rho(f(\mathcal{I}_v))$, we have $|\mathcal{A}_i \cap \mathcal{I}| = |\mathcal{A}_j \cap \mathcal{I}| = |\mathcal{I}_v \cap \mathcal{I}|$. As defined as a condition, $\mathcal{I}_v$ denotes informative features, thus we have $\mathcal{I}_v \subseteq \mathcal{I}$. Hence, $|\mathcal{I}_v \cap \mathcal{I}| = \mathcal{I}_v$.

## D    PROOF OF LEMMA 3.3

Here, we show the Lemma 3.3 again to avoid frequent turning pages for readers.

**Lemma.** *Given a set of features* $\mathcal{A}$ *and a set of informative features* $\mathcal{I}_v$*, if* $\rho(f(\mathcal{A})) = \rho(f(\mathcal{I}_v))$*, then we have* $\mathcal{I}_v \subseteq \mathcal{A}$*.*

**Proof:** Assume we have a set of features $\mathcal{A}$, and a set of non-informative features $\mathcal{A}_N \subseteq (\mathcal{F} \setminus \mathcal{I})$. We can have:

$$\begin{aligned}
(\mathcal{A} \cup \mathcal{A}_N) \cap \mathcal{I}_v &= (\mathcal{A} \cap \mathcal{I}_v) \cup (\mathcal{A}_N \cap \mathcal{I}_v) \\
&= (\mathcal{A} \cap \mathcal{I}_v) \cup \emptyset \\
&= \mathcal{A} \cap \mathcal{I}_v.
\end{aligned} \tag{2}$$

Hence, given a set $\mathcal{A}$, we can always find a larger set $\mathcal{A} \cup \mathcal{A}_N$ that satisfies $(\mathcal{A} \cup \mathcal{A}_N) \cap \mathcal{I}_v = \mathcal{A} \cap \mathcal{I}_v$. The $\mathcal{A} \cup \mathcal{A}_N$ can be defined as the new $\mathcal{A}$, then we always have $\mathcal{I}_v \subseteq \mathcal{A}$.

We show in this lemma that under $\rho(f(\mathcal{A})) = \rho(f(\mathcal{I}_v))$, the informative feature set $\mathcal{I}_v$ is always a subset of $\mathcal{A}$. Moreover, we show that $\mathcal{A}$ is only larger than $\mathcal{I}_v$, if $\mathcal{A}$ contains non-informative features. This implies that if we can somehow remove the non-informative features from $\mathcal{A}$, we can have $|\mathcal{A}| = |\mathcal{I}_v|$.

## E    FALSIFICATION EXPERIMENT CONFIGURATIONS

In this section, we show the experimental setup used for generating the results presented in Fig. 3 and Fig. 1.

For Fig. 3. We apply the *Constant* modification in Table 2 to GradCAM attribution maps on ImageNet. In addition, we set the noise level in ROAD to $0.01$ when performing noisy linear imputation. This noise level is also adopted in the experiments in Section 4.1.2, Section 4.2, and Section 4.3.

In the retraining experiment shown in Fig. 1, we use the CIFAR-10 (Krizhevsky et al., 2009a) dataset. The model is a tiny ResNet (He et al., 2016) with only 8 residual blocks. Training is conducted using Adam (Kingma & Ba, 2015) optimizer with a learning rate of $0.001$ and weight decay of $0.0001$. The batch size used for the training is $256$, and we train a model in $35$ epochs. Next, we describe how to construct the maliciously modified dataset for retraining. We generate a modified dataset from the original CIFAR-10 dataset. We perturb $5\%$ of each training image and replace the perturbed pixels with black pixels. The perturbation is class related. For different classes, we select different positions that are close to the edge of the image, so the object (which is usually at the center of the image) is barely removed.

## F    THE IMPLEMENTATION OF COMPLETENESS METRIC

For the completeness metric, the attribution threshold $t$ means that the pixels with attribution between $[t, 1]$ will be masked (i.e., value-based MoRF). We start from $v = 0.9$ and decrease $t$ by the step size of $0.1$. Algorithm 1 demonstrates the computation process.

## G    THE IMPLEMENTATION OF SOUNDNESS METRIC

For the soundness metric, the mask ratio $v$ means that the top $v$ pixels in an attribution map sorted in ascending order are masked (i.e., area-based LeRF). We start from $v = 0.98$ and decrease $v$ by

---

**Algorithm 1:** Completeness evaluation

---

**Input:** $f$: model; $\mathbb{D} = \{x_i, y_i, \mathcal{A}_i\}_{i=1}^N$: labeled dataset with input $x_i$, label $y_i$, and attribution map $\mathcal{A}_i$ for each $(x_i, y_i)$; $\phi$: perturbation function; $\psi$: noisy linear imputation function; $\mathbb{T} = \{0.9, 0.8, 0.7, \ldots, 0.1\}$: attribution thresholds; $Acc$: accuracy evaluation function.

**Output:** $\mathbb{S}_\Delta$: Accuracy differences associated with attribution thresholds $\mathbb{T}$.

Initialize $\tilde{\mathbb{S}}_\Delta$;
$s_0 \leftarrow Acc(f, \mathbb{D})$; // Accuracy on the unperturbed dataset
**foreach** $t$ *in* $\mathbb{T}$ **do**

> $\tilde{\mathbb{D}}_t \leftarrow \emptyset$; // Initialize dataset of imputed images
> **foreach** $x_i, y_i, \mathcal{A}_i$ *in* $\mathbb{D}$ **do**
>
> > $\hat{x}_i \leftarrow \phi(x_i, \mathcal{A}_i, t)$; // Perturb pixels whose attribution exceed $t$
> > $\tilde{x}_i \leftarrow \psi(\hat{x}_i)$; // Impute the perturbed image
> > $Append(\tilde{\mathbb{D}}_t, (\tilde{x}_i, y_i))$;
>
> $s_t \leftarrow Acc(f, \tilde{\mathbb{D}}_t)$; // Accuracy on the imputed dataset
> $Append(\mathbb{S}_\Delta, s_o - s_t)$; // Accuracy difference at the current step

**return** $\mathbb{S}_\Delta$;

---

the step size of $0.01$. This is equivalent to first inserting $2\%$ of the most important pixels in a blank canvas and inserting $1\%$ more pixels at each step. If the accuracy difference between the current step and the previous step is smaller than the threshold $0.01$, then the attribution of new added pixels is deemed to be false attribution and will be discarded. Algorithm 2 demonstrates the detailed procedure.

## H    VALIDATION AND BENCHMARK EXPERIMENTS

### H.1    IMAGENET IMAGES FOR FEATURE ATTRIBUTION

We randomly select 5 images for each class in the ImageNet validation set, obtaining a subset with 5000 images. When performing attribution, the images and attribution maps are resized to $224 \times 224$ before being fed into the pretrained VGG16 model.

### H.2    IMAGENET ATTRIBUTION MAPS

In this sub-section, we present the configurations for generating attribution maps on ImageNet. For GraCAM and IBA, We resize the resulting attribution maps to the same size as the corresponding input images. We use the implementations of GradCAM, DeepSHAP, IG in Captum (Kokhlikyan et al., 2020), and the implementation of ExPerturb in TorchRay [1], and the implementations of IBA and InputIBA in the official InputIBA codebase (Zhang et al., 2021). Some hyper-parameters for producing attribution maps are:

**GradCAM** We perform attribution on the `features.28` layer of VGG16 (i.e. the last convolutional layer).

**DeepSHAP and IG** We choose 0 as the baseline for attribution. We clamp the attribution to interval $[0, 1]$.

**ExPerturb** We set the area of perturbed region to 0.4.

**IBA and InputIBA** We attach the hidden information bottleneck to `features.17` layer of VGG16. We set the penalty coefficient $\beta_h = 20$ for the hidden information bottleneck, and $\beta_i = 20$ for the input information bottleneck.

In Fig. 7, we provide more visual results of different attribution methods. The samples are randomly selected.

---

[1] https://github.com/facebookresearch/TorchRay

---

**Algorithm 2:** Soundness evaluation

---

**Input:** $f$: model; $\mathbb{D} = \{x_i, y_i, \mathcal{A}_i\}_{i=1}^N$: labeled dataset with input $x_i$, label $y_i$, and attribution map $\mathcal{A}_i$ for each $(x_i, y_i)$; $\eta$: perturbation function; $\psi$: noisy linear imputation function; $\mathbb{M} = \{0.99, 0.98, 0.97, \ldots 0.01\}$: mask ratios (by area); $Acc$: accuracy evaluation function.;

$\epsilon$: accuracy threshold; $NewAdded$: function that identifies the new inserted features at the current step.

**Output:** $\mathbb{P}$: A set of tuples with each tuple being the accuracy and the attribution ratio.

Initialize $\mathbb{P}$;

$\hat{\mathcal{A}} \leftarrow \{\emptyset, \ldots, \emptyset\}_1^N$; // Initialize the set of features with false attribution for each sample.

$m_0 \leftarrow 1$; // Initialize the mask ratio at the previous step.

$s_0 \leftarrow 0$; // Initialize the accuracy at the previous step.

**foreach** $m$ *in* $\mathbb{M}$ **do**

$\quad$ $\tilde{\mathbb{D}}_m \leftarrow \emptyset$; // Initialize imputed dataset

$\quad$ **foreach** $x_i, y_i, \mathcal{A}_i$ *in* $\mathbb{D}$ **do**

$\quad\quad$ $\hat{x}_i \leftarrow \eta(x_i, \mathcal{A}_i, m)$; // Perturb image in LeRF order.

$\quad\quad$ $\tilde{x}_i \leftarrow \psi(\hat{x}_i)$; // Impute image.

$\quad\quad$ $\Delta\mathcal{A}_i \leftarrow NewAdded(\mathcal{A}_i, m, m_0)$; // Identify new added features

$\quad\quad$ $Append(\tilde{\mathbb{D}}_m, (\tilde{x}_i, y_i, \mathcal{A}_i, \Delta\mathcal{A}_i))$;

$\quad$ $s_m \leftarrow Acc(f, \tilde{\mathbb{D}}_m)$; // Accuracy on the imputed dataset.

$\quad$ **if** $s_m - s_0 < \epsilon$ **then**

$\quad\quad$ **foreach** $((\tilde{x}_i, y_i, \mathcal{A}_i, \Delta\mathcal{A}_i), \hat{\mathcal{A}}_i)$ *in* $(\tilde{\mathbb{D}}_m, \hat{\mathcal{A}})$ **do**

$\quad\quad\quad$ $\hat{\mathcal{A}}_i \leftarrow \hat{\mathcal{A}}_i \cup \Delta\mathcal{A}_i$; // Update the features with false attribution.

$\quad$ **foreach** $((\tilde{x}_i, y_i, \mathcal{A}_i, \Delta\mathcal{A}_i), \hat{\mathcal{A}}_i)$ *in* $(\tilde{\mathbb{D}}_m, \hat{\mathcal{A}})$ **do**

$\quad\quad$ $q_i \leftarrow \frac{|\mathcal{A}_i| - |\hat{\mathcal{A}}_i|}{|\mathcal{A}_i|}$;

$\quad$ $\bar{q} \leftarrow \frac{1}{N} \sum_{i=1}^N q_i$; // Attribution ratio at the current step.

$\quad$ $Append(\mathbb{P}, (s_m, \bar{q}))$; // Update results.

$\quad$ $s_0 \leftarrow s_m$; // Update the accuracy at the previous step.

$\quad$ $m_0 \leftarrow m$; // Update the mask ratio at the previous step.

**return** $\mathbb{P}$;

---

## H.3 VALIDATION TESTS

**Experiment on the synthetic dataset** We create a two-class dataset of 1000 sample data points, and each data point has 200 features. The data point is sampled from a 200-dimensional $\mathcal{N}(\mathbf{0}, \mathbf{I})$ Gaussian distribution. If all features for a sample point sum up to be greater than zero, we assign a positive class label to this sample. Otherwise, a negative class label is assigned (as described in the main text). The model is a linear model and can be formulated as $y = \sigma(\sum_i x_i)$, where $x_i$ is the $i$-th feature, and $\sigma(\cdot)$ is a step function that rises at $x = 0$, $\sigma(x) = -1$ if $x < 0$, and $\sigma(x) = 1$ if $x > 0$. In other words, the model also sums up all features of input and returns a positive value if the result is greater than zero. Hence, the model can classify the dataset with $100\%$ accuracy. Lastly, we describe how we create ground-truth attribution maps for this model and dataset. As the model is a linear model, and each feature $x_i$ is sampled from a zero-mean Gaussian distribution, the Shapley value for $x_i$ with $\sigma(x_i) = 1$ is then $1 \cdot (x_i - \mathbb{E}[x]) = x_i$. Similarly, the Shapley value for $x_i$ with $\sigma(x_i) = -1$ is $-x_i$. We confirm that attribution maps generated by Shapley values are fully correct for linear models. As a result, for positive samples, the attribution values are the same as feature values. For negative samples, the attribution values are the negation of feature values, which means that negative features actually contribute to the negative decision. Finally, we modify the attribution maps to be compatible with our soundness and completeness evaluation. Since our evaluation only supports positive attributions, we clip negative attribution values to zero. This conversion step has

Table 2: The modifications of attribution maps on ImagNet.

| Implementation | Removing attribution | Introducing attribution |
|---|---|---|
| Constant | Subtract the attribution map by a constant 0.6. | Add the attribution map with a constant 0.6. |
| Random | Sample a shift from $\mathcal{U}(-0.6, 0)$ for each pixel independently, and add the attribution of each pixel with the corresponding shift. | Sample a shift from $\mathcal{U}(0, 0.6)$ for each pixel independently, and add the attribution of each pixel with the corresponding shift. |
| Partial | Sort the attribution map in ascending order and select the pixels in indexing range $[0.6N, 0.8N]$, where $N$ is the number of pixels. Then set the attribution of these pixels to 0. | Sort the attribution map in ascending order and select the pixels in indexing range $[0, 0.4N]$, where $N$ is the number of pixels. Then set the attribution of these pixels to $q_{0.8}$, where $q_t$ denote the $t$-th quantile of attribution values. |

no negative effect on the actual evaluation. The rest of the evaluation setup is identical to other experiments.

**The experiment on the ImageNet**  We implement three different modification schemes and modify the GradCAM attribution maps. The details thereof are presented in Table 2. We show some examples of modified attribution maps in Fig. 8. The images are randomly selected.

### H.4  ENSEMBLE METHODS OF IG

In each ensemble method, we use an isotropic Gaussian kernel $\mathcal{N}(\mathbf{0}, 0.3 \cdot \mathbf{I})$ to sample 20 noisy samples for a given input sample. Fig. 9 shows additional examples of IG and its ensemble methods. The images are randomly selected.

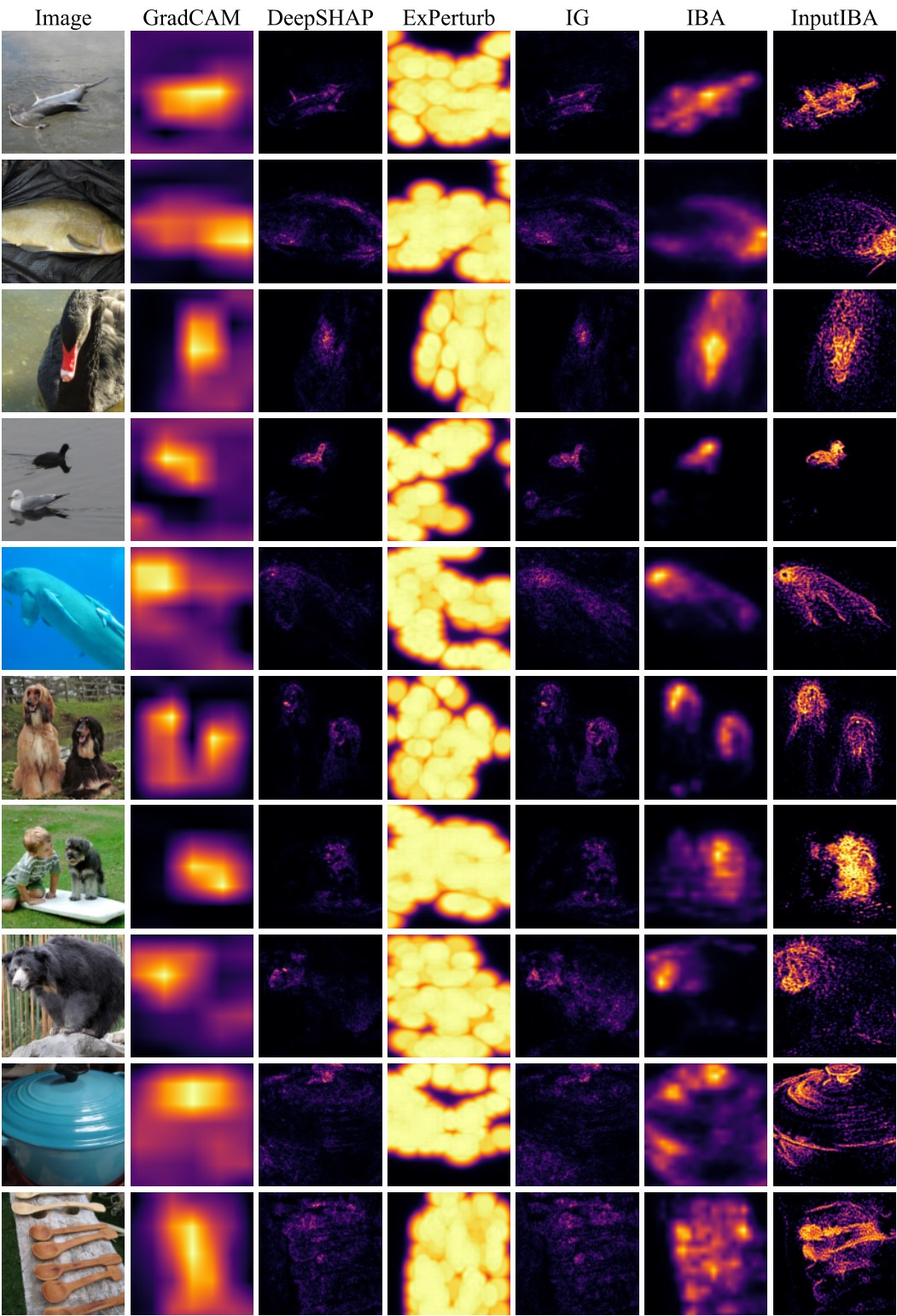

Figure 7: Examples of various attribution maps. The authors ensure that samples are not cherry-picked.

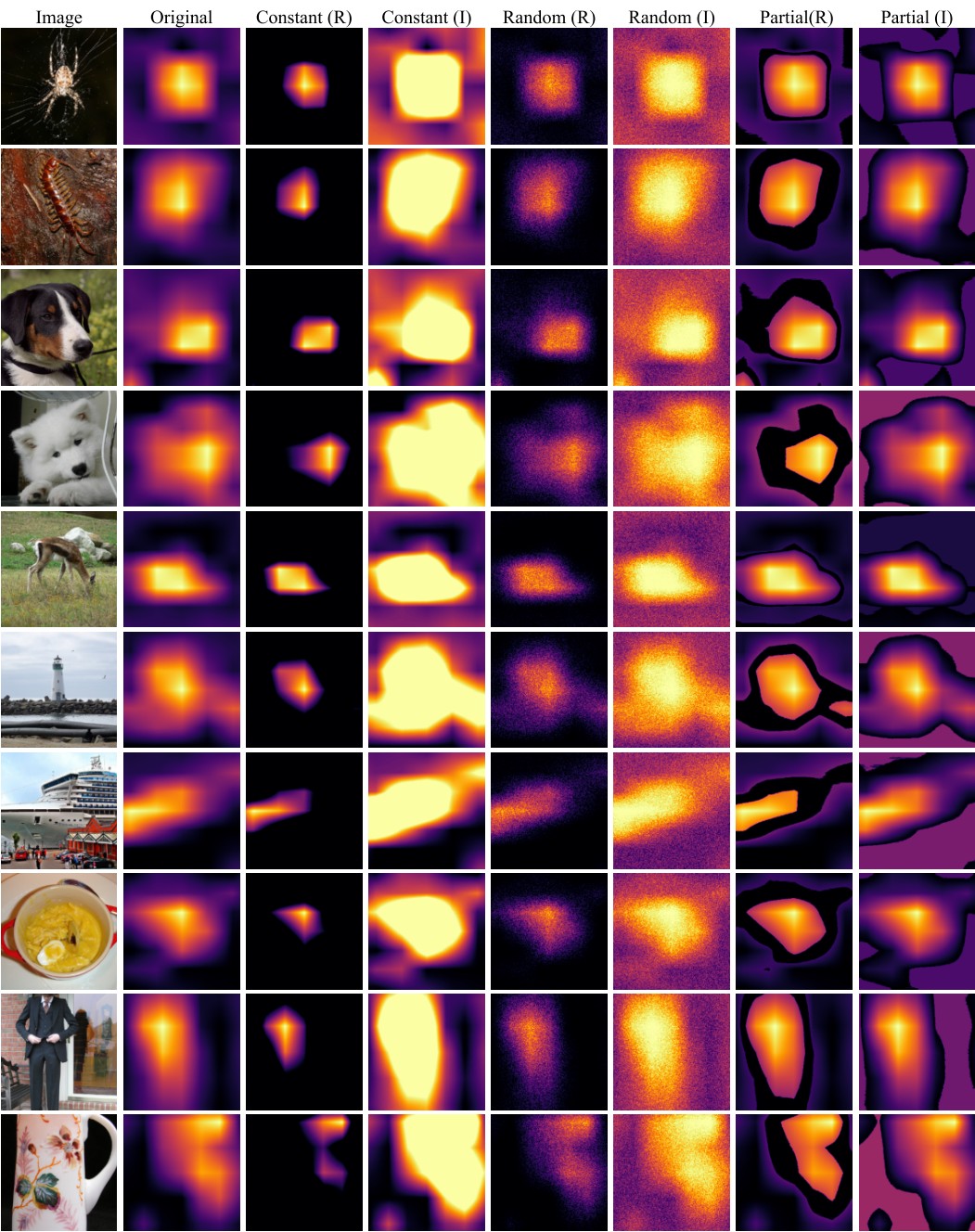

Figure 8: Examples of modified attribution maps. (R) and (I) denote the *Remove* and *Introduce* modification, respectively. The authors ensure that samples are not cherry-picked.

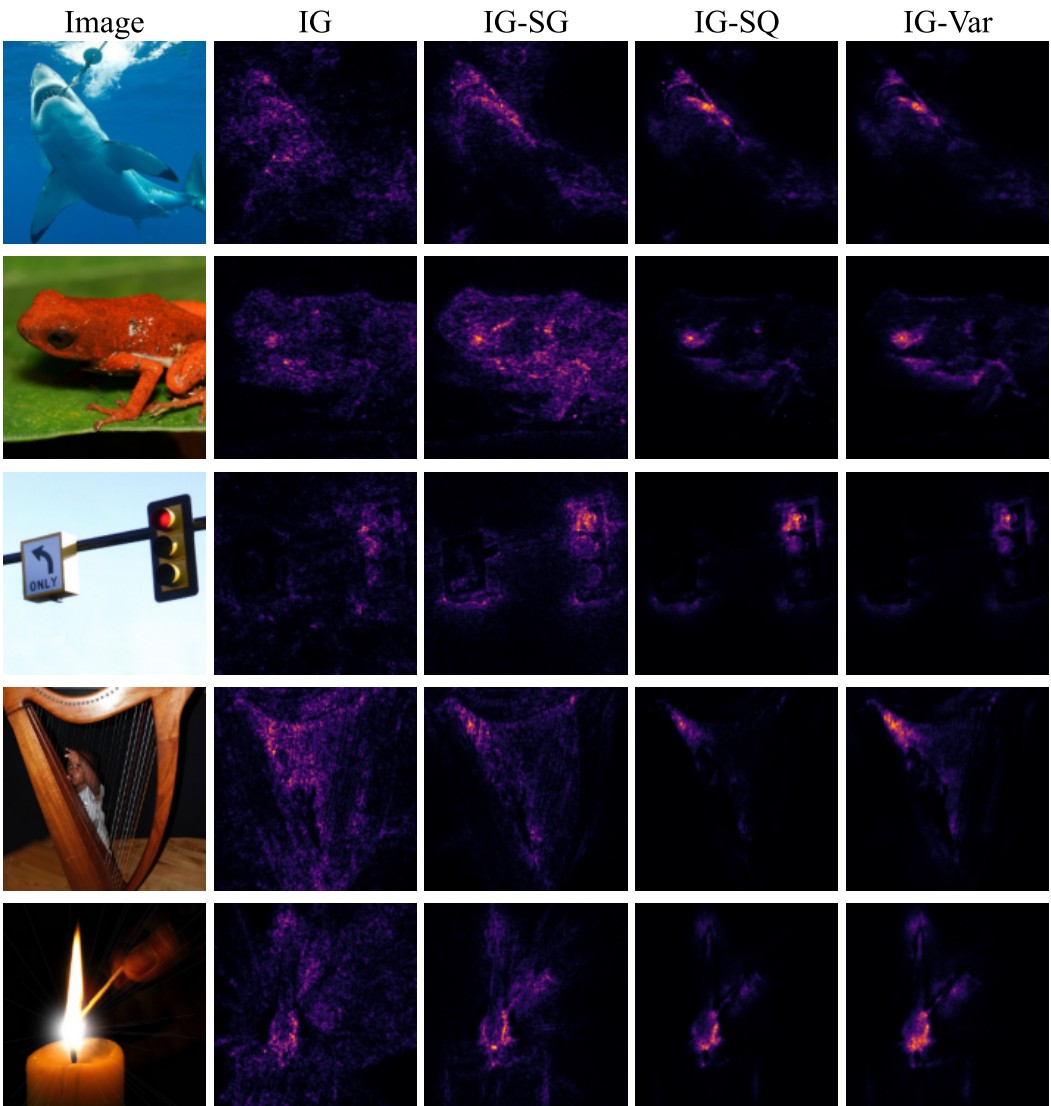

Figure 9: Examples of IG and ensembled attribution maps. The authors ensure that samples are not cherry-picked.

