# OpenReview forum: "Soundness and Completeness: An Algorithmic Perspective on Evaluation of Feature Attribution"
_ICLR.cc/2023/Conference — Submitted to ICLR 2023_

### Official Review · Reviewer_rKgk · 2022-10-23

**Confidence:** 4
**Correctness:** 2
**Technical Novelty And Significance:** 1
**Empirical Novelty And Significance:** 1
**Recommendation:** 3

**Clarity, Quality, Novelty And Reproducibility:**

The method has limited novelty, and the relationship with prior work could be improved.

**Strength And Weaknesses:**

--- Strengths ---

- Several flaws about some existing evaluations are pointed out.

--- Weaknesses ---

About the method:
- One of the key challenges with the method is that we don't have access to the ground-truth important features. The authors thus resort to approximate approaches, which seem geared primarily at comparing two attribution vectors rather than scoring one in isolation. Furthermore, the final procedure involves re-doing the prediction repeatedly with different amounts of (un)important features held out. Ultimately, this procedure is extremely similar to insertion/deletion (Petsiuk et al, 2018), but perhaps with a couple extra steps. The novelty thus seems quite limited.
- The ideas behind soundness and completeness are very simple, there's no need to resort to vague notions of "algorithm theory." As a reader it's a bit frustrating to see "algorithm theory" mentioned multiple times and then find out what the authors actually mean by it. You can just say what this method is: testing if the explanation identifies the important features but nothing more.
- Describing other methods, the authors write: "While prior works about attribution method evaluations mainly focus on explaining whether a method is correct or not, they ignore the fact that violation of either soundness or completeness can result in a partially correct algorithm." I don't get the point of this, it's not like other methods return a 0/1 value of whether the attribution is correct - they also return scores. What's the significant difference?
- Theorem 3.3 is a simple result, perhaps it should be re-named as a proposition?
- Mixing the ideas of set cardinality and summation within a set seems a bit sloppy. For example, in the proof for theorem 3.1, $|I \setminus A| = |I \setminus (A \cap I)|$ is a valid equality if we're talking about set cardinality, but not if we're talking about summation.
- When talking about the "information level", the notation $I_v$ defined as $I_v \subseteq I$ s.t. $|I_v| = v$ doesn't make sense to me. There can be many subsets of that size, so which one is it?
- In theorem 3.4, what is the relationship between $\mathcal{A}^*$ and $\mathcal{A}$? Should we have $\mathcal{A}^* \subseteq \mathcal{A}$? If so, this should be specified.

About discussions of prior work:
- In describing prior work, the authors write that one issue is duplicate or conflicting definitions (e.g., "sensitivity"). I agree that it's an issue, but by the authors' account it seems like an issue with terminology rather than the metrics themselves, which is perhaps not as interesting.
- In describing prior work, the authors frequently refer to "order-based" and "model-retraining" as the two groups of existing methods. This categorization of metrics doesn't make sense to me, they're overlapping categories and some methods belong to neither. For example, ROAR (Hooker et al, 2019) is both order-based and involves model retraining. Sensitivity-n (Ancona et al, 2018) is neither. This seems like a strange way to group existing evaluation methods.
- The cost of model retraining is indeed a problem, and this is a complex issue because naive approaches like replacing with zeros are problematic in their own way (see Hooker et al, 2019). However, the authors don't actually "solve" this problem, they just impute the missing features. I'm not sure this can be described as a novel solution, as it's the same approach used by many other methods already (e.g., Petsiuk et al 2018, Ancona et al 2018).
- The authors make a point of criticizing methods that are only sensitive to ordering, but what about the methods that aren't? For example, sensitivity-n (Ancona et al, 2018, already cited) and faithfulness [1]. These test the specific attribution scores rather than just their ordering.

[1] Bhatt et al, "Evaluating and aggregating feature-based model explanations" (2021)

**Summary Of The Paper:**

This paper is written to introduce two notions of evaluation for model explanations: soundness and completeness. The definitions for these metrics are straightforward but require prior knowledge of the ground-truth important features. The authors therefore suggest practical alternatives, which appear to reduce to procedures very similar to existing ones: making predictions with different subsets of (un)important features and analyzing how the predictions change.

**Summary Of The Review:**

The authors propose two new criteria for evaluating model explanations. They are intended to address shortcomings in existing methods but re-use many of their key ideas. Thus, the novelty and significance are limited.

---

> ### Author Response · Authors · 2022-11-16
> **Reply (3) to rKgk**
>
> > In describing prior work, the authors write that one issue is duplicate or conflicting definitions (e.g., "sensitivity"). I agree that it's an issue, but by the authors' account it seems like an issue with terminology rather than the metrics themselves, which is perhaps not as interesting.
>
> The reviewers pointed to the Introductory section. We also agree that this is an issue more related to *terminology*. Also, that’s the reason why we say this point only in the introduction section. The incentive of this statement is to raise consensus among readers that there is an urgent need for a well-defined, uniform structure for evaluation, and this work is an attempt towards this goal. We never mention this terminology issue anywhere except in the Introduction section since it is technologically not interesting and novel.
>
> > In describing prior work, the authors frequently refer to "order-based" and "model-retraining" as the two groups of existing methods. This categorization of metrics doesn't make sense to me, they're overlapping categories and some methods belong to neither. For example, ROAR (Hooker et al, 2019) is both order-based and involves model retraining. Sensitivity-n (Ancona et al, 2018) is neither. This seems like a strange way to group existing evaluation methods.
>
> In Section 2 of our paper, we explicitly categorize evaluation metrics into expert-grounded metrics and functional-grounded metrics. Section 3.1 only shows that two types of operations can cause issues. One is order-based evaluation, and the other is retraining-based evaluation. Here, we only identify them as the causes of different issues. We never categorize evaluation metrics into these two types throughout our work. If a metric uses both operations (like ROAR), it will suffer from both issues mentioned in our work.
>
> > The cost of model retraining is indeed a problem, and this is a complex issue because naive approaches like replacing with zeros are problematic in their own way (see Hooker et al, 2019). However, the authors don't actually "solve" this problem, they just impute the missing features. I'm not sure this can be described as a novel solution, as it's the same approach used by many other methods already (e.g., Petsiuk et al 2018, Ancona et al 2018).
>
> In this work, we never discuss imputation in our methodology. We only apply imputation as one building block of our implementation. We use imputation because prior work has shown that imputation is essential to mitigate unintended effects. We acknowledge prior work’s contribution and adopt imputation in our work. However, we don’t mention imputation as our contribution or novel solution. The contributions in this work are:
> * Identify issues with prior evaluation metrics and empirically show issues with experiments.
> * Design an evaluation framework based on soundness and completeness in the absence of ground truth, and avoid retraining.
> * Design validation experiments to show the correctness of our metric. In the revised version, we also add comparative experiments to show the advantage of our metric.
>
> We would appreciate it if the reviewer could point out where exactly we mentioned imputation as our contribution so that we can make further clarification.
>
> > The authors make a point of criticizing methods that are only sensitive to ordering, but what about the methods that aren't? For example, sensitivity-n (Ancona et al, 2018, already cited) and faithfulness [1]. These test the specific attribution scores rather than just their ordering.
>
> Although Sensitivity-N is not order-based, it is still scale-invariant. This is because it uses correlation as an evaluation outcome. If we shift attribution values uniformly or multiply all values with a constant, the correlation will not change. This is a perfect example showing the prevalent phenomenon that many feature attribution evaluations are value-insensitive. Still, there are few evaluations that are not solely relying on ordering. For instance, Sanity Check works by perturbing model parameters. But a majority of evaluation metrics are only sensitive to ordering.

---

> ### Author Response · Authors · 2022-11-16
> **Reply (2) to rKgk**
>
> > Describing other methods, the authors write: "While prior works about attribution method evaluations mainly focus on explaining whether a method is correct or not, they ignore the fact that violation of either soundness or completeness can result in a partially correct algorithm." I don't get the point of this, it's not like other methods return a 0/1 value of whether the attribution is correct - they also return scores. What's the significant difference?
>
> We thank the reviewer for pointing this out. The phrasing we used had ambiguity, which somehow delivered an impression that prior work only returns a binary result. Hence, we have changed  “whether a method is correct or not” to “the relative correctness of a method”.
>
> > Theorem 3.3 is a simple result, perhaps it should be re-named as a proposition?
>
> Lemma 3.3, and Lemma 3.2, are intermediate results for deriving Theorem 3.4. Hence, they are named Lemmas.
>
> > Mixing the ideas of set cardinality and summation within a set seems a bit sloppy. For example, in the proof for theorem 3.1, $|\mathcal{I} \setminus \mathcal{A}|=|\mathcal{I} \setminus (\mathcal{A} \setminus \mathcal{I})|$ is valid equality if we're talking about set cardinality, but not if we're talking about summation.
>
> For the sake of simplicity, we overload the operator $|\cdot|$. This operator computes the total attribution or information. In the above equation, the $|\cdot|$ only applies to the outer scope. That said, $\mathcal{I} \setminus (\mathcal{A} \setminus \mathcal{I})$ is the pure logical computation of sets. After that, the $|\cdot|$ computes the contained information
>
> > When talking about the "information level", the notation $\mathcal{I}_v$ defined as $\mathcal{I}_v \subseteq \mathcal{I}$ s.t. $|\mathcal{I}_v| = v$ doesn't make sense to me. There can be many subsets of that size, so which one is it?
>
> We thank the reviewer for raising this issue. $\mathcal{I}_v$ is indeed not a unique set. We are more interested in the fact that $\mathcal{I}_v$ contains informative features whose information adds up to information level $v$. The value $v$ is not calculated in our work, instead, we use model performance to indicate the information level based on Assumption 1. Nevertheless, we need a unique $\mathcal{I}_v$ to ensure consistent evaluation of Soundness. The way to constrain the set $\mathcal{I}_v$ is through the process of expanding $\mathcal{I}_v$. In our implementation, we start the inclusion with the most salient features. Hence, set $\mathcal{I}_v$ contains informative features that have higher attribution than other informative features not included in $\mathcal{I}_v$.
> We did not include this additional definition in the original version of our work. In the revised version, we add a statement in Section 3.2.2 to clarify that $\mathcal{I}_v$ contains only the most salient features compared with other features that are not in $\mathcal{I}_v$.
> We would like to state that this modification does not violate our original implementation. All evaluation results are still based on the definition of $\mathcal{I}_v$.
> > In theorem 3.4, what is the relationship between $\mathcal{A}^*$  and $\mathcal{A}$? Should we have $\mathcal{A}^* \subseteq \mathcal{A}$? If so, this should be specified.
>
> We thank the reviewer for pointing this detail out. The relation between $\mathcal{A}^*$ and $\mathcal{A}$ is as the reviewer stated. We have specified this relationship in Theorem 3.4 for better clarity.

---

> ### Author Response · Authors · 2022-11-16
> **Reply (1) to rKgk**
>
> We thank the reviewer for constructive comments and suggestions.
>
> > One of the key challenges with the method is that we don't have access to the ground-truth important features. The authors thus resort to approximate approaches, which seem geared primarily at comparing two attribution vectors rather than scoring one in isolation. Furthermore, the final procedure involves re-doing the prediction repeatedly with different amounts of (un)important features held out. Ultimately, this procedure is extremely similar to insertion/deletion (Petsiuk et al, 2018), but perhaps with a couple extra steps. The novelty thus seems quite limited.
>
> Progress perturbation forms a general framework for many evaluation metrics in feature attribution. We also follow this framework. However, we note that many metrics like Insertion/Deletion and ROAD only consider the order of features. Therefore, these metrics cannot distinguish between two attribution maps that arrange the features in the same order but contain different values. In contrast, our proposed metrics do not suffer from this problem. We have added an experiment in Fig. 3c to compare Completeness and ROAD directly.
>
> In addition, the formulation of Completeness/Soundness is unique and has never been explored by prior work in attribution methods. These two metrics measure the entirely different properties of attribution methods than Insertion/Deletion do. As noted in our paper, Insertion/Deletion checks whether the order of features is correct. However, Completeness/Soundness measure to what extent attribution values match the ground truth.
>
>
>
> > The ideas behind soundness and completeness are very simple, there's no need to resort to vague notions of "algorithm theory." As a reader it's a bit frustrating to see "algorithm theory" mentioned multiple times and then find out what the authors actually mean by it. You can just say what this method is: testing if the explanation identifies the important features but nothing more.
>
> The reviewer mentioned that our method is “testing if the explanation identifies the important features but nothing more”. We would like to clarify that, as discussed in our paper, if an evaluation metric only tests whether the explanation identifies important features, then this metric cannot give us informative feedback. For example, an explanation method can satisfy the metric defined by the reviewer by always returning important features, but only returning a small portion of the entire important features. Or an explanation method that can identify all important features but also identify many non-important features as important. In both cases, the explanation method is not optimal. This is why we introduce definitions from algorithm theory in this work. In algorithm theory, a correct algorithm is always sound and complete. If either soundness or completeness is violated, the algorithm is not correct. In this work, our idea is to focus on the fact that attribution methods are also algorithms so that we can apply evaluation metrics for algorithms on attribution methods. The intuition of introducing the term “algorithm theory” is to make a clear linkage.

---

> > ### Comment · Reviewer_rKgk · 2022-11-27
> > **Author response**
> >
> > Thanks to the authors for writing a detailed response to the reviewer concerns. Based on the response, I'm not inclined to raise my score. My main concern is that the proposed methods provide limited benefit over existing approaches, but I also believe the work could be improved by addressing the writing quality (e.g., removing any mention of "algorithm theory"), discussions of prior work (sensitivity-n, faithfulness, ROAD), and the experiments section (more clearly demonstrating that these metrics are better). About the imputation topic, because the authors don't propose anything new here it would also make sense to reduce the focus on model-retraining methods, which are discussed prominently in the introduction and section 3.1.

---

### Official Review · Reviewer_7Hq4 · 2022-10-23

**Confidence:** 3
**Correctness:** 4
**Technical Novelty And Significance:** 2
**Empirical Novelty And Significance:** 2
**Recommendation:** 5

**Clarity, Quality, Novelty And Reproducibility:**

-
The definitions of soundness and completeness are not somehow explained clearly in the submission and is confusing since the metrics depend on each other. It would be good to improve this.

**Strength And Weaknesses:**

+
Overall I think evaluating explanations is an important topic and new metrics/techniques to help evaluate explanations would be very useful to the community.

-
The significance of these two metrics is not clear based from the submission. What is the risk of not computing these metrics?
There are new pieces of work based on computing ground truth: https://arxiv.org/pdf/2104.14403.pdf
It would be useful to show empirically if the proposed new metrics could become a good substitute for evaluating explanations without the need for ground truth, at least to some extent.

It would be good to show the results for tabular and text data.


**Summary Of The Paper:**

Authors propose two new metrics to evaluate feature attribution explanations: completeness and soundness which are based on algorithm theory. Authors show limitations of several existing order-based and model-retraining based metrics. The metrics are evaluated on several explainability methods such as GradCAM, DeepSHAP, IG, etc. for images.

**Summary Of The Review:**

See above

---

> ### Author Response · Authors · 2022-11-15
> **Reply to Reviewer 7Hq4**
>
> We thank the reviewer for constructive comments and suggestions.
>
> > The significance of these two metrics is not clear based from the submission. What is the risk of not computing these metrics? There are new pieces of work based on computing ground truth: https://arxiv.org/pdf/2104.14403.pdf It would be useful to show empirically if the proposed new metrics could become a good substitute for evaluating explanations without the need for ground truth, at least to some extent.
>
> We explain the risk by showing the limitations of prior metrics.
> * Many metrics, such as Insertion/Deletion and ROAD [1], only measure if features are correctly ordered w.r.t. attribution value. It first removes features according to two different orders (MoRF/LeRF) and then checks the change in accuracy. Our Completeness/Soundness overcomes this limitation. **We have added a new experiment in the revised paper (See Fig. 3b and Fig. 3c) to compare completeness and ROAD directly**. Furthermore, the information returned from ROAD is more limited than Completeness/Soundness. The latter metrics evaluate attribution methods from different perspectives, providing us with more insights into attribution methods’ limitations and potential directions for improvement. As shown in Section 4.3, we can know in which direction the ensemble methods improve over their baseline.
>
> * Zhou et.al. [2] advocate for injecting ground truth features that correlate with re-assigned labels. This approach has a problem:  **learning from the semi-natural dataset deviates from the original task**. Consequently, the model retrained on the semi-natural dataset is much simpler than the original model. When explaining these two models using the same attribution method, the attribution method can attain quite different performances. Therefore, we cannot use the evaluation results on the retrained model to represent the performance of attribution methods on real-world datasets. To demonstrate this issue, we have appended a new experiment in Section 3.1. We first train a model respectively on a real-world dataset, a semi-natural dataset, and a pure synthetic dataset, obtaining three trained models. Then, we use ROAD to evaluate attribution methods on the three models. Surprisingly, GradCAM outperforms IG and DeepSHAP on the model trained on the real-world dataset, while IG and DeepSHAP are much better than GradCAM on the model trained on the semi-natural and pure synthetic dataset. We show the results in Appendix B. Due to the above reason, we do not adopt the model retraining or ground-truth injection approaches.
>
>
>
> > It would be good to show the results for tabular and text data.
>
> The synthetic data in our validation experiments are discrete tabular data. We use the synthetic data to verify the correctness of our metric, while many prior methods don’t provide such validation in their works. In conclusion, our experiments cover both tabular and vision data. We omit the experiments on text data because: (1) attribution methods on recent NLP models, such as transformers, are out of scope for this work; (2) imputation methods for tokens are barely explored, so there is no proper solution for solving the OOD issue for text data.
>
> -------------------------------------------------------------------------------------------------------------------
>
> [1] Yao Rong, Tobias Leemann, Vadim Borisov, Gjergji Kasneci, and Enkelejda Kasneci. A consistent and efficient evaluation strategy for attribution methods. In International Conference on Machine Learning, pp. 18770–18795. PMLR, 2022.
>
> [2] Zhou, Yilun, et al. "Do feature attribution methods correctly attribute features?." Proceedings of the AAAI Conference on Artificial Intelligence. Vol. 36. No. 9. 2022.

---

### Official Review · Reviewer_iZ2R · 2022-10-23

**Confidence:** 3
**Correctness:** 2
**Technical Novelty And Significance:** 2
**Empirical Novelty And Significance:** 2
**Recommendation:** 3

**Clarity, Quality, Novelty And Reproducibility:**

Clarity: I do have quite some complaints on the writing, mainly in Sec. 3. As a result, I could not exactly follow the exposition of the idea.

Novelty: many ideas in this paper have been seen in the literature, such as correlating model prediction/performance (e.g. $\rho$) with feature importance, and using imputation to solve OOD/leakage issues.

Reproducibility: the attached code seems to be of high quality, so I believe that reproduction of the work would be easy.

**Strength And Weaknesses:**

**Section 1 and 2**

While I agree with the authors on their motivation to make evaluations that are not order-based or require model-retraining, I believe that there are some misrepresentations of the prior work, in particular (Hooker et al. 2019) and (Zhou et al. 2022) which I happen to be quite familiar with.

> (Zhou et al. 2022) assumes that a retrained model only learns watermarks added into the original images of a semi-nature dataset

From my understanding, they reassigned the labels to be only correlated with the injected features (e.g. watermarks). While this is kind of an extreme operation, this does guarantee that none of the other features are actually correlated with the label, and the model could not use any other features to achieve better than random chance performance.

> Many evaluation metrics are only order-sensitive, meaning that they only evaluate whether one feature is more important than another while ignoring how differently the two features contribute to the output.

The attr% metric used by (Zhou et al. 2022) does depend on the actual value of the attribution.

> We use the modified dataset for retraining and report the accuracy on the unperturbed test set to evaluate how many remaining features are learned during retraining.

This is in reference to (Hooker et al. 2019) paper on the discussion of retraining. Here, the authors used the perturbed data in training and unperturbed data in testing, creating a distribution shift, which is exactly what this paper argued to avoid, by introducing the retraining step (perturbing both training and test data). Thus, I fail to see what the results actually demonstrate, because if the model (retrained or not) is tested on a different distribution, its performance can be affected by many OOD issues.

**Section 3**

I have many questions on it.

Assumption 1: $\rho$ is defined as a performance metric (e.g. accuracy). However, there can be features that have high impact on model prediction, but not in a way that contributes to the model performance. For example, a feature could improve the performance on a subset of inputs and hurt that on another subset, balancing out the $\rho$ metric. This is related to "Model behavior" in Fig. 1 of [1].

Sec. 3.2.2: I don't quite understand the definition of $I_v$ and information level. Do we assume an ordering on $I$, such that $I_v$ is the top-$v$ important features in $I$, or something else?

Related to the point above, I guess throughout, we assume that the saliency map generates a binary split of features into $A$ and $F\backslash A$. Most attributions are continuous. I think the authors used some thresholding, which may be related to the information level, but I don't find the details.

The authors mentioned that $I$ is unknown (i.e. the unknown ground-truth problem). However, reading through Sec. 3, I still don't understand how $I$ is in practice inferred or approximated. I would appreciate a high-level overview and roadmap at the beginning of this section before going into technical definitions and proofs.

Minor: Def 3.1 and 3.2 can be written as $I\subseteq A$ and $A\subseteq I$.

[1] https://arxiv.org/pdf/2011.14878.pdf

**Section 4**

The experiment covers a wide range of explanation methods. However, at the end of the day, all conclusions are drawn based on the two proposed metrics. Their relationship with other metrics is not clear. In other words, since these two metrics are again *defined* (starting from Def. 3.1, 3.2 and Assumption 1), it is not clear how or why they are a better alternative to existing approaches, and what their limitations are.

**Summary Of The Paper:**

This paper proposed two new metrics, completeness and soundness, for evaluating attribution maps. The two most important characteristics of these two metrics are that they do not require model retraining and are sensitive to the actual values of the attribution scores, not just their ordering. These two metrics are evaluated for many popular saliency maps, and several conclusions are drawn.

**Summary Of The Review:**

Overall, I do not believe that this paper makes a sufficiently significant contribution to merit acceptance. However, I would also want to note that the non-clarity of writing (in my opinion) prevents me from giving a very confident assessment.

---

> ### Author Response · Authors · 2022-11-16
> **Reply (3) to Reviewer iZ2R**
>
> > many ideas in this paper have been seen in the literature, such as correlating model prediction/performance (e.g. $ρ$) with feature importance, and using imputation to solve OOD/leakage issues.
>
> To the best of our knowledge, using model performance as the indicator for feature attribution evaluation is the de-facto approach. Imputation is not the contribution of our work. In our paper, we don’t discuss imputation in our proposed metrics. We only mention imputation in the experiment section to show that we follow recent works and apply imputation to fix the OOD issue. One of the contributions of our work is to identify and demonstrate the existing issues of prior evaluation approaches. Based on the observation, we designed two metrics that overcome these issues. Moreover, the proposed metrics allow us to evaluate how the alignment between attribution and important features differs in different ways (e.g., soundness and completeness). This refined evaluation metric can help the community better identify the limitations of current attribution methods and provide directions for improvements, as we have shown in Section 4 with benchmarking attribution methods.

---

> ### Author Response · Authors · 2022-11-16
> **Reply (2) to Reviewer iZ2R**
>
> > Sec. 3.2.2: I don't quite understand the definition of $I_v$ and information level. Do we assume an ordering on $I$, such that $I_v$ is the top-$v$ important feature in $I$, or something else?
>
> $\mathcal{I}_v$ is a set with features that have in total $v$ information. The information is a variable measuring the importance of these features for *decision-making*. Hence, we can use model performance to compare information levels indirectly. In our implementation, we calculate Soundness by expanding the set with features with the highest attribution. In the end, $\mathcal{I}_v$ has two properties. Firstly, $\mathcal{I}_v$ is the set of informative features with total information reaching information level $v$. Secondly, informative features included in $\mathcal{I}_v$ have the highest attributions among all informative features. Suppose we have a mapping function $g$ that maps model accuracy to the information level. In this case, we have $\mathcal{I}_v$ as the set of informative features that can achieve $c$ model accuracy, where $g(c) = v$. This shows we never compute $v$ but use model accuracy as an indicator for a specific information level.
>
> In practice, we use the above definition to calculate soundness at the different information levels. Here, we use Lemma 3.2, which says if two sets of features can lead to the same model accuracy, then the information level of these two feature sets is the same.
>
> > Related to the point above, I guess throughout, we assume that the saliency map generates a binary split of features into $\mathcal{A}$ and $\mathcal{F} \setminus \mathcal{A}$. Most attributions are continuous. I think the authors used some thresholding, which may be related to the information level, but I don't find the details.
>
> Since attribution maps are continuous, we proposed relative soundness and completeness metrics based on the definition of sound and complete attribution methods to tackle the compatibility problem with continuity. For completeness, we threshold features based on their attribution values. For soundness, we threshold the information levels based on model accuracy. These thresholds are reflected in our soundness and completeness plots, in which we have attribution value and accuracy on the respective $x$-axis. We have also revised the paper to have more implementation details in Sections 3.2.1 and 3.2.2.
>
> > The authors mentioned that $I$ is unknown (i.e. the unknown ground-truth problem). However, reading through Sec. 3, I still don't understand how $I$ is in practice inferred or approximated. I would appreciate a high-level overview and roadmap at the beginning of this section before going into technical definitions and proofs.
>
> We note that $\mathcal{A}$ contains a proportion of features that truly contribute to model performance. This part of features is defined as the $\mathcal{I}_v$ with a constraint that the information sums up to $v$. However, we still cannot observe or measure $v$ directly. In our paper, we resort to using model performance as an indirect indicator to determine whether $\mathcal{I}_v$ is contained in the input. In practice, we progressively expand $\mathcal{A}$ by include new features. At each step, we examine if the newly added features are truly informative by checking the change in model performance. This additional examining process determines whether the newly added features belong to $\mathcal{I}_v$.
>
> We value the suggestion from the reviewer and have added more details at the beginning of Section 3.2.2 to show that $\mathcal{I}_v$ is inferred by looking at model performance.
>
>
> > Minor: Def 3.1 and 3.2 can be written as $I \subseteq A$ and $A \subseteq I$.
>
> We thank the reviewer for suggesting this. This suggestion is indeed equivalent to our formulation in the paper. Our formulation emphasizes the fact that both sets $\mathcal{I}$ and $\mathcal{A}$ are the subset of the feature set $\mathcal{F}$. The original formulation also emphasizes that features are elements during comparison.
>
> ### Results and comparison
> > The experiment covers a wide range of explanation methods. However, at the end of the day, all conclusions are drawn based on the two proposed metrics. Their relationship with other metrics is not clear. In other words, since these two metrics are again defined (starting from Def. 3.1, 3.2 and Assumption 1), it is not clear how or why they are a better alternative to existing approaches, and what their limitations are.
>
> We have updated Fig. 3c and added a comparison to ROAD. In 2 out of 3 modification schemes, ROAD fails to distinguish the modified attribution maps. This experiment shows that our proposed completeness can effectively overcome the limitation of order-based attribution methods.

---

> ### Author Response · Authors · 2022-11-16
> **Reply (1) to Reviewer iZ2R**
>
> We thank the reviewer for the comments and suggested improvements.
>
> > From my understanding, they reassigned the labels to be only correlated with the injected features (e.g. watermarks). While this is kind of an extreme operation, this does guarantee that none of the other features are actually correlated with the label, and the model could not use any other features to achieve better than random chance performance.
>
> We admit that the label reassignment decorrelates the original image features from new labels. However, learning from the semi-natural dataset deviates from the original task. Consequently, the model retrained on the semi-natural dataset is much simpler than the original model. When explaining these two models using the same attribution method, the attribution method can attain quite different performances. Therefore, we cannot use the evaluation results on the retrained model to represent the performance of attribution methods on real-world datasets. To demonstrate this issue, we have appended a new experiment in Section 3.1. We first trained a model respectively on a real-world dataset, a semi-natural dataset, and a pure synthetic dataset, obtaining three trained models. Then, we use ROAD to evaluate attribution methods on the three models. Surprisingly, GradCAM outperforms IG and DeepSHAP on the model trained on the real-world dataset, while IG and DeepSHAP are much better than GradCAM on the model trained on the semi-natural and pure synthetic dataset. We plot the result in Appendix B.
>
> > The attr% metric used by (Zhou et al. 2022) does depend on the actual value of the attribution.
>
> We agree that the attr% metric depends on attribution values. The purpose of mentioning (Zhou et. al. 2022) in the paper is to demonstrate the problems associated with modified dataset. As noted in Section 3.1 and above, (Zhou et. al. 2022) has inherent limitations.
>
>
> > Here, the authors used the perturbed data in training and unperturbed data in testing, creating a distribution shift, which is exactly what this paper argued to avoid, by introducing the retraining step (perturbing both training and test data).
>
> We thank the reviewer for pointing out the implementation details of ROAR [1]. To summarize, we intentionally use an unperturbed test set to show how features from the original dataset are (not) learned by the retrained model. We know the potential adverse effect of distribution shift when using an unperturbed test set. To best mitigate the effect, we perturb only a tiny portion of every training image, and the perturbation is at the edge of every training image. Subsequently, the object in the original learning task is barely affected. The evaluation results show a significant performance drop on the unperturbed test set. This result objects to the statement that the “retraining model learns on remaining features,” which is the principle assumption in some retraining-based evaluations.
>
>
> > Assumption 1: $\rho$ is defined as a performance metric (e.g. accuracy). However, there can be features that have high impact on model prediction, but not in a way that contributes to the model performance.
>
> We aim to study the systematic behavior of feature attribution methods. For every input image, the procedure of obtaining the attribution map is the same, which is the attribution algorithm itself. In our setting, an attribution value only indicates a positive effect on decision-making. If a feature has an inconsistent influence on the model’s performance on different subsets, then the feature is not considered a positive factor for overall decision-making. Hence, it should not receive a high attribution value.
>
> ---
> [1] Sara Hooker, Dumitru Erhan, Pieter-Jan Kindermans, and Been Kim. A benchmark for interpretability methods in deep neural networks. Advances in neural information processing systems, 32, 2019.

---

### Official Review · Reviewer_4ijn · 2022-10-27

**Confidence:** 4
**Correctness:** 3
**Technical Novelty And Significance:** 2
**Empirical Novelty And Significance:** 1
**Recommendation:** 5

**Clarity, Quality, Novelty And Reproducibility:**

Quite clear, fair quality, some novelty, and should be reproducible using the code in the supplementary material.

**Strength And Weaknesses:**

The problem of evaluating saliency method is a pertinent one and several ways have been proposed in recent literature approaching from varying perspectives. The proposal in this paper is quite sensible, paralleling the well understood notion of precision and recall (in case of soundness and completeness respectively). However, I had a few concerns:

1. While the framing is unique, it appears to me that the proposed metrics have strong relationship to the insertion and deletion based metrics. For example, to my best understanding, completeness is essentially the deletion metric with the only difference that instead of ordered removal (remove top x% attributed pixels), it uses absolute thresholds (remove pixels with attribution greater than y) before measuring the performance change. Did I miss something? What’s the justification for introducing new terminology if the change is indeed that minor?

2. How does this proposal differ from ROAD other than using values instead of order as noted above?

3. I find the connection between the formalization and the actual implementation a little tenuous. While considerable space is devoted to the theory, the actual algorithms are delegated to the appendices. I also found it difficult to see how the theory truly informed the algorithms.

4. The results section is somewhat lacking. The experiments on the toy dataset conform to the expectation which is good but not enough evidence of the goodness of the metrics. Then the section proceeds to benchmark the attribution methods using the proposed metrics. While interesting, the goal of the results section should be to convince the reader of the merit of the proposal. Just applying the metric to existing techniques doesn’t build any confidence in me for the work in the paper. Any insights thus generated have a confound of whether the observations say something about the proposed metrics or about the techniques studied.

5. Minor: “semi-nature datasets”, “nature constraints”. The formatting style of I in equation (1) and proof of Theorem 3.4 is inconsistent in places.

**Summary Of The Paper:**

In this paper, the author propose an evaluation method for feature attribution based on the concept of soundness and completeness. The two notions are described along with how they might be approximated when ground truth feature importance is unavailable. The results show the metrics to behave sensibly when saliency maps are manipulated in specific ways.

**Summary Of The Review:**

Overall, while there is some merit to the submission in its attempt to use formal and standardized way for evaluating attribution methods, I am not entirely convinced of its novelty beyond the introduction of new terminology. The writing is fairly clear but the details on actual implementation are not present in the paper’s main body. Based on these observations, I am not inclined for the paper to be accepted at the venue.

---

> ### Author Response · Authors · 2022-11-15
> **Reply to Reviewer 4ijn (1/2)**
>
> We thank the reviewer for the useful comments and suggested improvements.
>
> ### Reply to point 1:
> First, we emphasize that *Insertion* and *Deletion* focus on the same property of saliency maps: the order of features in terms of their saliency values. More specifically, **Insertion measures the deviation from the optimal ascending order**, while **Deletion measures the deviation from the optimal descending order**. ROAD [1] has proposed to measure the consistency of evaluation results with MoRF (Most important Remove First) and LeRF (Least important Remove First). This proposal suggests that insertion and deletion measure the same property in different directions.
>
> The reviewer mentioned that Completeness is similar to Insertion/Deletion in that only feature removal is changed from order-based to value-based. It is to note that the change follows our definitions and theorems rather than pure heuristics. For our completeness metric, the relative Completeness is calculated based on different attribution values, as we define the attribution set $\mathcal{A}$ as features with attributions above a certain level. Hence, we designed to remove features based on their attribution values and measure the model performance (here is the model's accuracy) as a metric of $1 - completeness$. In addition, the Soundness metric is an entirely new notation. In our work, we predefine a set of different accuracy levels and then gradually include attributed features until the model reaches these accuracy levels. During this process, we also use model accuracy to determine if the included features are informative. Finally, we calculate the ratio of included informative features to the total included features as Soundness. The difference can be seen in the plot reporting Soundness in our paper. In the Soundness metric, we plot a curve of attribution ratio (y-axis) vs. accuracy (x-axis). However, Insertion/Deletion plots a curve of accuracy (y-axis) vs. mask ratio (x-axis). The introduction of the attribution ratio is unique.
> Regarding measuring the performance of feature attribution methods, the optimal order of features is usually measured by order-based methods like Insertion/Deletion, ROAR, and ROAD. Various causes can hurt this optimal order. For instance, increasing the attribution of non-informative features or decreasing the attribution of informative features can lead to the order of features being different from the optimal order. However, Insertion/Deletion cannot distinguish these two cases, while Soundness and Completeness can.
>
> ---
> ### Reply to point 2:
>
> Like Insertion/Deletion, ROAD still only measures the order of features. It removes features according to two different orders (MoRF/LeRF) and then checks the change in accuracy. Our Completeness/Soundness differs from ROAD in the following aspects:
> * Completeness/Soundness overcome the limitation of ROAD. Like Insertion/Deletion, ROAD only measures if features are correctly ordered w.r.t. attribution value. Completeness/Soundness do not suffer from this issue. We have added a new experiment in the revised paper (see Fig. 3b and Fig. 3c) to compare completeness and ROAD directly. In addition, as noted in the previous point, Soundness applies a different approach. Soundness computes the attribution ratios at different accuracy levels, while ROAD reports accuracies at different mask ratios.
> * The information returned from ROAD is more limited than Completeness/Soundness. The latter metrics evaluate attribution methods from different perspectives, providing us with more insights into attribution methods’ limitations and potential directions for improvement. As shown in Section 4.3, we can know in which direction the ensemble methods improve over their baseline.
>
> ---
> ### Reply to point 3:
>
> We thank the reviewer for raising this concern. In the main text, we would like to emphasize the theoretical part, specifically, how to define feature sets, define the notion of Soundness/Completeness, and measure Soundness/Completeness without information about ground truth. Moreover, another significant contribution of this work is empirically showing issues of previous evaluation approaches. We draw your attention to *Appendix F and Appendix G*, where we describe the implementation of the metric and the detail of the algorithm. In the last paragraph of Section 3.2.1, we briefly describe how we implement Completeness.
>
> To better link the theory and the implementation, we have added more words in Section 3.2.2 about the information level to help readers gain more intuition. Furthermore, according to your suggestion, we have revised the last paragraph in Section 3.2.2 to provide more details on the implementation of Soundness.
>
>
>
> [1] Yao Rong, Tobias Leemann, Vadim Borisov, Gjergji Kasneci, and Enkelejda Kasneci. A consistent and efficient evaluation strategy for attribution methods. In International Conference on Machine Learning, pp. 18770–18795. PMLR, 2022.

---

> > ### Author Response · Authors · 2022-11-15
> > **Reply to Reviewer 4ijn (2/2)**
> >
> >
> > ### Reply to point 4:
> >
> > In this work, we have validation experiments on our proposed methods. The validation experiments use purposefully designed attribution maps to validate our metrics. Moreover, we have added comparative results in the revised paper to show the advantage of our metric.
> > We have updated Fig. 3c and added a comparison to ROAD. In 2 out of 3 modification schemes, ROAD fails to distinguish the modified attribution maps. This experiment shows that our proposed completeness can effectively overcome the limitation of order-based attribution methods. We only include ROAD as the baseline in this comparative experiment, as it is the most recent evaluation approach that adopts order-based perturbation. Other order-based methods have the same issue revealed in this work and would achieve no better result in our validation experiment. For retraining-based methods, we use Fig. 1 to show the issue with retraining, and Appendix B shows the problem with the modified dataset. Our proposed metric has no retraining in the loop to avoid the issue with retraining.
> >
> > We also want to highlight the difficulty of “evaluating evaluation metrics.” It is harder to evaluate metrics under one unified framework than evaluate methods under one evaluation metric. We can design attribution maps to compare order-based with non-order-based evaluation metrics. However, for metrics with retraining in the loop, we cannot design a fair experiment to compare them with non-retraining metrics directly.
> >
> > ---
> > ### Reply to point 5:
> >
> > We thank the reviewer for pointing out the language issues. We have updated the manuscript and solved these issues.
> >
> > ---
> > ### Reply to other points:
> > >  …I am not entirely convinced of its novelty beyond the introduction of new terminology.
> >
> > In this work, our contribution is beyond introducing new terminology. We would like to emphasize our contribution here:
> > * We reveal several issues in existing evaluation metrics and designed experiments to show these issues.
> > * We design new evaluation frameworks for feature attribution. Our proposals are strictly based on definitions and derived theories, not causal heuristics. Our metrics do not have the issues we revealed. We also empirically show the correctness of our metric, as well as comparative results with prior metrics.
> > * We benchmark feature attribution methods with our metrics. The results on attribution methods are novel and show more insights into the properties of evaluated attribution methods.

---

### Author Response · Authors · 2022-11-18
**General Response**

We thank the reviewers for their helpful feedback and constructive suggestions. We have individually responded to their comments. Based on the discussion below, we have revised our manuscript to incorporate their suggestions.


Specifically, we have made the following changes:

* As suggested by reviews, we add a comparative experiment on ImageNet to show the difference between our proposed metrics and ROAD (the most recent order-based evaluation metric) in Fig. 3.
* Remove old Fig. 1 and the descriptive text of Fig. 1 from the revised paper to avoid duplication. The content has been moved together with the additional result to Fig. 3 in the revised paper.
* Add a paragraph in Section 3.1 to discuss the issue of *Zhou et. al.* [1]. Multiple reviewers have asked whether *Zhou et. al.* is better for attribution evaluation, so we add this part to address their concerns. The corresponding experiment results are added in Appendix B due to the page limit in the main text.
* We revise Section 3.2.2 to provide more explanation about the information level and how we measure it. We also add more implementation details about the algorithm.
* Merge the result of benchmarking attribution methods and the result of benchmarking ensemble methods into one plot (Fig. 5) in the revised version to enhance readability.
* Minor changes suggested by reviewers (e.g. “semi-nature” to “semi-natural”).

---
[1] Yilun Zhou, Serena Booth, Marco Tulio Ribeiro, and Julie Shah. Do feature attribution methods correctly attribute features? In Proceedings of the AAAI Conference on Artificial Intelligence, volume 36, pp. 9623–9633, 2022.

---

### Decision · Program_Chairs · 2023-01-20

**Decision:**

Reject

**Justification For Why Not Higher Score:**

Limited novelty.

**Justification For Why Not Lower Score:**

N/A

**Metareview: Summary, Strengths And Weaknesses:**

The paper aims to propose a new evaluation method to measure the soundness and completeness of feature attribution methods. Although this is an important task, the proposed evaluation seems very similar to the existing ones in the literature, so all the reviewers think the novelty is insufficient for ICLR. Further, there are some other issues including the experiments and the clarity.